# LEARN TO REASON EFFICIENTLY WITH ADAPTIVE LENGTH-BASED REWARD SHAPING

**Wei Liu**[1]* **Ruochen Zhou**[2] **Yiyun Deng**[1] **Yuzhen Huang**[1] **Junteng Liu**[1]
**Yuntian Deng**[3] **Yizhe Zhang**[4] **Junxian He**[1]*
[1]The Hong Kong University of Science and Technology  [2]City University of Hong Kong
[3]University of Waterloo  [4]Apple

## ABSTRACT

Large Reasoning Models (LRMs) have shown remarkable capabilities in solving complex problems through reinforcement learning (RL), particularly by generating long reasoning traces. However, these extended outputs often exhibit substantial redundancy, which limits the efficiency of LRMs. In this paper, we investigate RL-based approaches to promote reasoning efficiency. Specifically, we first present a unified framework that formulates various efficient reasoning methods through the lens of length-based reward shaping. Building on this perspective, we propose a novel **L**ength-b**A**sed **S**t**E**p **R**eward shaping method (LASER), which employs a step function as the reward based on target length. LASER surpasses previous methods, achieving a superior trade-off between performance and efficiency. Next, we further extend LASER based on two key intuitions: (1) The reasoning behavior of the model evolves dynamically during training, necessitating reward specifications that are also adaptive and dynamic; (2) Rather than uniformly encouraging shorter or longer chains of thought (CoT), we posit that length-based reward shaping should be difficulty-aware i.e., it should penalize lengthy CoTs more for easy queries. This approach is expected to facilitate a combination of fast and slow thinking, leading to a better overall tradeoff. The resulting method is termed LASER-D (Dynamic and Difficulty-aware). Experiments on five open-weight models from 1.5B to 32B demonstrate that our approach significantly enhances both reasoning performance and response length efficiency. For instance, LASER-D achieves a $+5.3$ improvement on AIME2024 while reducing token usage by **64%**. Further analysis reveals that our RL-based compression produces more concise reasoning patterns with less redundant "self-reflections". All resources (Models, Code, Data) are available at https://github.com/hkust-nlp/Laser.

## 1 INTRODUCTION

Recent advancements leveraging reinforcement learning (RL) (DeepSeek-AI et al., 2025; Kimi et al., 2025; OpenAI et al., 2024; Zeng et al., 2025) demonstrate that LLMs can evolve into powerful Large Reasoning Models (LRMs), capable of producing extended chains of thought (CoT) to enhance their reasoning abilities. However, these longer reasoning trajectories come at the cost of increased token usage and potentially incorporate more compounding errors. Many of the generated tokens tend to be unnecessarily verbose. For example, LRMs may output thousands of tokens to solve elementary math problems that could otherwise be addressed within just a few hundred tokens, as shown in Figure 1 (right). This phenomenon is commonly referred to as the *over-thinking* issue (Chen et al., 2025).

Recently, RL-based CoT compression methods have demonstrated the most favorable balance between token efficiency and accuracy (Kimi et al., 2025; Arora & Zanette, 2025; Aggarwal & Welleck, 2025; Hou et al., 2025; Fang et al., 2025; Xu et al., 2025). These methods can be broadly categorized into three classes as shown in Table 1: *budget* (Aggarwal & Welleck, 2025; Xu et al., 2025; Qi et al., 2025), *adaptive* (Fang et al., 2025; Tu et al., 2025), and *full* (Kimi et al., 2025; Hao et al., 2024; Wu et al., 2025). *Budget* constrains the model to reason within a fixed response or thinking length. *Adaptive* leverages a dual-mode design that switches between thinking and non-thinking (or

---

*Correspondence to Wei Liu (wliucn@cse.ust.hk) and Junxian He (junxianh@cse.ust.hk)

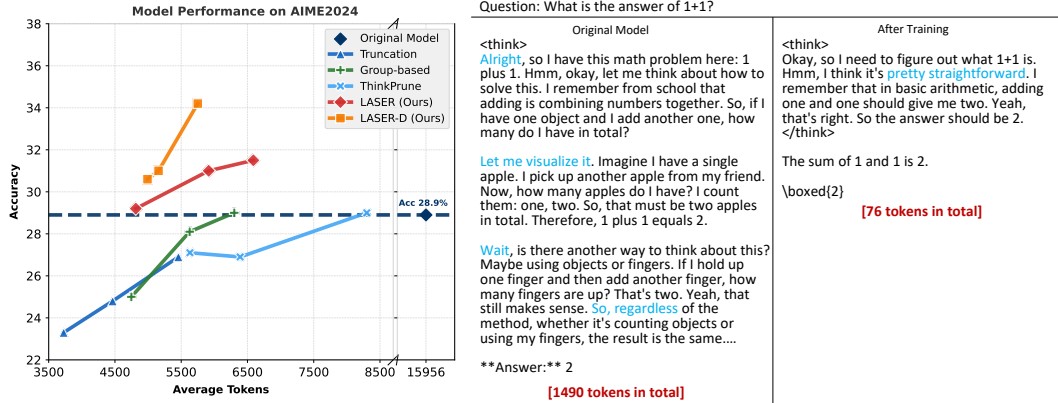

Figure 1: **Left**: Accuracy and response length on AIME2024. For the figure of more benchmarks, please refer to Appendix B. Each point represents a single training run with different hyper-parameters. Given the high computational cost of obtaining this figure, the base model used is DeepSeek-R1-Distill-Qwen-1.5B. Results on 7B and 32B models are in §5.3. Our methods, LASER and LASER-D achieve the best efficacy-efficiency trade-off compared to all other methods. Notably, they yield a $+5.3$ improvement in accuracy while reducing tokens usage by $64\%$ compared to the original model. **Right**: Example of a reasoning process after LASER-D training. While the original model produces meaningless "self-reflection" repeatedly for trivial questions like "1+1=?", LRMs after LASER-D training recognize such questions during thinking and provide the answer directly.

long- and short-thinking) for different queries to improve efficiency. *Full* pursues policy models that can achieve extreme efficiency across all difficulty levels and problems, while simultaneously striving to preserve, or even improve, reasoning accuracy. These different designs naturally lead to different trade-offs. Models trained with *budget*-mode methods require a manually defined budget, but such human-crafted budgets are inherently suboptimal—often too restrictive for hard questions and overly generous for easy ones. While *adaptive* mode teaches models to decide when to think, in practice, they tend to collapse into a degenerate pattern: skipping thinking on simple questions but still expending redundant reasoning effort on harder ones. By contrast, models trained under the *full* mode reason significantly more efficiently even on difficult problems. Yet achieving accuracy improvements alongside such compression remains challenging. As shown in Table 1, most prior methods across all three modes struggle to simultaneously deliver substantial token reductions on hard problems and consistent accuracy gains. Some even rely on multi-stage pipelines with additional SFT, but still fall short of this goal.

In this paper, we follow the line of *full* CoT compression, aiming to improve reasoning efficiency and accuracy through concise, principled single-stage methods. We begin with a simple baseline in *full* mode that yields surprisingly effective results (§2). Specifically, we further train long CoT reasoning models using RL with a rule-based correctness reward, while restricting the context window size to a smaller value than the model's typical generated length, so that long responses will be truncated. This approach can substantially improve token efficiency with only a modest reduction in accuracy. To better understand this truncation method and to connect it with other RL approaches that incorporate length-based rewards (Kimi et al., 2025; Arora & Zanette, 2025; Aggarwal & Welleck, 2025), we introduce a unified length-based reward shaping perspective that encompasses various RL strategies for mitigating overthinking (§3). Building on this reward shaping formulation, we recognize the limitations of previous length-based reward methods and propose a novel reward shaping approach (§3.3) that employs a step function guided by a desired target length. We refer to this approach as LASER (**L**ength-b**A**sed **S**t**E**p **R**eward). LASER achieves the best trade-off between reasoning performance and efficiency among all evaluated baselines, delivering consistent gains in both reasoning efficiency and accuracy.

Next, we identify two key points that are lacking in LASER: (1) the desired reasoning length should evolve during training as the model's reasoning behaviors dynamically change, and (2) rather than uniformly encouraging short or long CoT, length-based reward shaping should be *difficulty-aware* – allowing harder questions a higher token limit while constraining easier questions to fewer tokens. To this end, we propose a **D**ynamic and **D**ifficulty-aware **L**ength-b**A**sed **S**t**E**p **R**eward for RL (LASER-D), which adaptively applies different length reward shaping based on problem difficulty. Notably, our algorithm is fully automated with an integrated automatic adapting module, eliminating the need for manual procedural tuning.

Table 1: Comparison of RL-based CoT compression methods. *Compression mode*: *Full* aims to learn policy models that improve efficiency across all questions; *Budget* constrains the model to follow a target response or thinking length. *Adaptive* trains the model to switch between thinking and non-thinking (or long- and short-thinking) modes. *Single stage* indicates whether the method can be completed within a single-stage RL training pipeline without additional SFT or multi-stage RL. *Hard problems efficiency* reflects whether the method enables models to achieve more than $50\%$ token reduction on AIME24. *Accuracy improvements* indicates whether the method improves reasoning accuracy. *Difficulty-aware* shows whether the method explicitly accounts for varying reasoning effort across different question difficulty levels.

| Methods | Compression Mode | Single Stage | Hard Problems Efficiency | Accuracy Improvements | Difficulty Aware |
|---|---|---|---|---|---|
| ThinkPrune (Hou et al., 2025) | Full | ✗ | ✓ | ✗ | ✗ |
| Kimi-k1.5 (Kimi et al., 2025) | Full | ✓ | ✓ | ✗ | ✗ |
| Efficient Reasoning (Arora & Zanette, 2025) | Full | ✓ | ✓ | ✗ | ✗ |
| L1 (Aggarwal & Welleck, 2025) | Budget | ✓ | ✓ | ✗ | ✗ |
| E1 (Xu et al., 2025) | Budget | ✓ | ✓ | ✗ | ✗ |
| AnytimeReasoner (Qi et al., 2025) | Budget | ✓ | ✗ | ✓ | ✗ |
| Thinkless (Fang et al., 2025) | Adaptive | ✗ | ✗ | ✗ | ✓ |
| AutoThink (Tu et al., 2025) | Adaptive | ✗ | ✗ | ✓ | ✓ |
| LAPO (Wu et al., 2025) | Full | ✗ | ✗ | ✓ | ✓ |
| LASER | Full | ✓ | ✓ | ✓ | ✗ |
| LASER-D | Full | ✓ | ✓ | ✓ | ✓ |

Table 2: Results of baseline truncation method with different context window. $T_k$ denotes the models after RL training with context window $k$. Accuracy (%) with average token usage for each dataset. "Original" denotes the original DeepSeek-R1-Distill-Qwen-1.5B.

| | Accuracy (%) | | | | | Generation Length (tokens) | | | | |
|---|---|---|---|---|---|---|---|---|---|---|
| | MATH 500 | AIME | AMC | Olympiad Bench | Avg. | MATH 500 | AIME | AMC | Olympiad Bench | Avg. |
| Original | 83.9 | 28.9 | 71.6 | 43.3 | 56.9 | 5042 | 15956 | 8202 | 11510 | 10177 |
| $T_{8192}$ | 81.8 | 24.8 | 70.9 | 43.9 | 55.3 | 1795 | 4465 | 2560 | 2841 | 2915 |
| $T_{6144}$ | 80.9 | 20.2 | 66.2 | 42.1 | 52.3 | 1351 | 2821 | 1917 | 1947 | 2009 |
| $T_{4096}$ | 77.7 | 19.2 | 62.2 | 38.5 | 49.4 | 1054 | 2481 | 1484 | 1564 | 1646 |

We conduct comprehensive experiments on five reasoning models ranging from 1.5B to 32B parameters, across four challenging benchmarks: MATH500, AIME2024, AMC2023, and OlympiadBench. Our extensive evaluations demonstrate that our proposed LASER series outperforms existing works, while LASER-D achieves the best efficacy-efficiency trade-off. Unlike methods that improve token efficiency at the expense of accuracy, our proposed approaches deliver substantial gains in both dimensions—even on the challenging AIME2024 benchmark. For example, applying LASER-D to DeepSeek-R1-Distill-Qwen-1.5B improves accuracy by $+\mathbf{5.3}$ percentage points while reducing token usage by $\mathbf{64}\%$ on AIME24.

## 2 TRUNCATION: A SIMPLE YET EFFECTIVE BASELINE

In this section, we a simple yet effective *full*-mode baseline, where we simply set max generation length to a value significantly smaller than the model's original context window during RL training (e.g., 8,192 vs. 32,768 tokens in DeepSeek-R1-Distilled). Responses exceeding this limit are truncated and treated as incorrect, encouraging concise yet accurate reasoning. This baseline has also been explored recently in concurrent works Hou et al. (2025); Luo et al. (2025). In our experiments, we adopt DeepSeek-R1-Distill-Qwen-1.5B as the base model and investigate the effects of truncation by limiting maximum generation lengths to 4,096, 6,144, and 8,192 tokens.

**Effectiveness of Truncation** Table 2 presents the performance of models across various benchmarks under different truncation sizes. Compared to the original model, surprisingly, RL training with a context window of 8192 tokens achieves a substantial $\mathbf{71}\%$ improvement in token efficiency, while maintaining competitive accuracy with a 1.6 absolute point degradation on average. This demonstrates that truncation is a simple yet effective approach for enhancing reasoning efficiency in LRMs.

**Efficacy-Efficiency Trade-off** Although truncation improves efficiency overall, it disproportionately harms harder tasks. On AIME, accuracy drops by $\mathbf{4.1}$ under an 8192-token limit and by $\mathbf{9.7}$ at 4096, compared to only $\mathbf{7}\%$ on MATH500. This stems from the higher truncation ratio on harder benchmarks: over $\mathbf{75}\%$ of AIME responses exceed 8192 tokens, versus $\mathbf{15}\%$ for MATH500 (Figure 7). These results highlight the efficacy–efficiency trade-off: truncation reduces tokens effectively,

but the accuracy cost is amplified on complex tasks that demand longer reasoning. Next, we formulate the truncation baseline from the reward shaping perspective and connect it with related works.

## 3 A Unified View on Truncation and Length-based Rewards

In this section, we aim to understand the truncation baseline and other single-stage RL-based efficient reasoning approaches of the *full* compression mode through a unified perspective. We first connect them together via length-based reward shaping, and then derive new alternatives with this view.

### 3.1 The Unified Formulation

Here we first present a unified formulation, and then we show how the truncation baseline and other works fit into this formulation. Specifically, we define the reward function with two parts: a correctness term $C(y)$ and a length-based term $S(y)$ controlled by a control variable $\lambda(y)$:

$$\hat{R}(x, y) = C(y) + \lambda(y) \cdot S(y) \tag{1}$$

In most length reward methods, $C(y) = R(x, y)$, representing the original rule-based reward for correctness. However, in truncation-based approaches, $C(y) = 0$ as we discuss below. The term $S(y)$ denotes the length reward, which varies across different methods.

**Formulating the Truncation Baseline**   As shown in Table 3, truncation is a special case of the length reward with $C(y) = 0$, where the target length $L_T$ is enforced by the context window. ThinkPrune (Hou et al., 2025) is another truncation-based approach, which extends vanilla truncation by introducing an adaptive target length $L_A$ to replace fixed target length $L_T$. They iteratively choose $L_A$ and separate their training into three stages. Table 3 also outlines other formulations, we will introduce them individually in the following sections.

### 3.2 Connecting Previous Efforts Together

In this part, we build on our unified formulation in Eq. 1 and aim to connect previous approaches, below we describe several main categories of them. Table 3 formulates different length-based reward shaping by different designs for $C(y)$, $\lambda(y)$ and $S(y)$. Parameter $\alpha$ is a hyperparameter coefficient that controls the magnitude of the length reward $S(y)$. We provide detailed explanations for each formulation in Appendix H.

**Group-based Reward**   In group-based reward, the length reward $S(y)$ is designed to encourage brevity by assigning higher scores to shorter responses within a rollout group, such as Efficient Reasoning (Arora & Zanette, 2025) and Kimi-k1.5 (Kimi et al., 2025) as formulated in Table 3. However, this comparison-based approach can lead to reward hacking. Models tend to exploit $S(y)$ by generating overly concise responses, particularly for simpler questions. We demonstrate this phenomenon for the Efficient Reasoning baseline in Figure 8a and Figure 8b in Appendix E, where training accuracy initially decreases while total reward increases. Additionally, Table 4 shows a more significant drop in MATH500 accuracy compared to other methods, further supporting this observation.

**Budget-based Reward**   Although budget-based reward methods are typically categorized under the *budget* compression mode rather than *full*, some concise single-stage approaches (Aggarwal & Welleck, 2025) can still be unified into our formulation since they are also implemented through length-based reward shaping. Budget-based rewards use query-specific target lengths (budgets) and penalize responses that deviate from these instructions. While this mitigates reward hacking seen in group-based schemes, it can destabilize training. Models require exposure to diverse budgets, but in large context windows (e.g., 16,384 tokens), these targets become sparsely distributed, causing reward fluctuations. Figure 8b illustrates this instability. Using L1-Max as a representative method, we observe that with smaller contexts (4,096 tokens), it achieves stable reward increases comparable to other methods. However, with 16,384-token contexts, rewards fluctuate significantly and underperform alternative approaches.

Table 3: Formulation of different approaches based on Eq. 1. $C(y)$ is mainly for correctness, $S(y)$ is the length reward, and $\lambda(y)$ is a control variable to control how length reward is applied. $\mathbb{I}(R)$ stands for $\mathbb{I}(R(x,y)=1)$ and $\mathbb{I}(\cdot)$ is an indicator function. $\rho$ is the negative reward given for incorrect responses. $L(y)$ is the length of the generated response. $\alpha$ is the coefficient that controls the magnitude of the length reward. The shapes of different rewards are shown in the visualization, where x axis is the length of the response. Blue represents the curve for correct responses, while Red represents the curve for incorrect responses. For approaches, ThinkPrune, LASER-D, there are different lines with similar colors indicate that the reward is dynamic which is realized by different $L_A$ values. The details of visualization are available in the Appendix M.

| Name | $C(y), \lambda(y)$ | $S(y)$ | Visualization |
|---|---|---|---|
| **Truncation Method** | | | |
| Vanilla Truncation | $0, 1$ | $\begin{cases} R(x,y) & \text{if } L(y) \leq L_T \\ \rho & \text{if } L(y) > L_T \end{cases}$ | |
| ThinkPrune (Hou et al., 2025) | $0, 1$ | $\begin{cases} R(x,y) & \text{if } L(y) \leq L_A \\ \rho & \text{if } L(y) > L_A \end{cases}$ | |
| **Group-based Reward** | | | |
| Efficient Reasoning (Arora & Zanette, 2025) | $R(x,y), \mathbb{I}(R)$ | $-\alpha \cdot \sigma\left(\frac{L(y) - \text{Mean}(y)}{\text{STD}(L)}\right)$ | |
| Kimi-k1.5 (Kimi et al., 2025) | $R(x,y), 1$ | $\begin{cases} 0.5 - \frac{L(y) - L_{\min}}{L_{\max} - L_{\min}} & \text{if } \mathbb{I}(R) = 1 \\ \min\left(0, \ 0.5 - \frac{L(y) - L_{\min}}{L_{\max} - L_{\min}}\right) & \text{if } \mathbb{I}(R) = 0 \end{cases}$ | |
| **Budget-based Reward** | | | |
| L1-Exact (Aggarwal & Welleck, 2025) | $R(x,y), 1$ | $-\alpha \cdot |L(y) - L_T|$ | |
| L1-Max (Aggarwal & Welleck, 2025) | $0, \mathbb{I}(R)$ | $\text{clip}(\alpha \cdot (L(y) - L_T) + \delta, 0, 1)$ | |
| **Length-Based Step Reward and Variants** | | | |
| LASER | $R(x,y), \mathbb{I}(R)$ | $\alpha \cdot \mathbb{I}(L(y) \leq L_T)$ | |
| LASER-D | $R(x,y), \mathbb{I}(R)$ | $\alpha \cdot \mathbb{I}(L(y) \leq L_A)$ | |

## 3.3 BRIDGING THE GAP: LENGTH-BASED STEP REWARD

As shown in Eq. 1 and visualized in Table 3, a key limitation of the truncation method is that it assigns the same penalties to overlong responses as it does to incorrect ones, which may over-penalize long but correct explorations. To address this issue, we extend it as a novel reward shaping approach called **L**ength-b**A**sed **StE**p **R**eward (LASER), which adopts a step reward function guided by a target length, rather than performing hard truncation.

Specifically, we design the length reward term $S(y)$ as an indicator function based on a target length $L_T$. This function assigns a length-based bonus to responses shorter than $L_T$. We also set the context window significantly larger than the target length $L_T$ (e.g., 16,384 vs. 4,096) where truncation rarely happens. And the length reward term $S(y)$ is only activated when responses are correct, thereby improving the efficacy-efficiency trade-off. As visualized in Table 3, LASER closely resembles the vanilla truncation approach; the only difference is that, instead of truncating long responses, it awards bonus rewards to correct responses that do not exceed the target length. To balance the correctness reward $C(y)$ and length reward $S(y)$, we follow a typical setting and set $\alpha$ as 0.5. We also find that LASER is robust across different values of $\alpha$ (see §5.2).

Empirical results are demonstrated in Figure 1 and Table 4, training with the LASER reward achieves improved efficacy-efficiency trade-off compared to all previous methods. Notably, it is the first approach to simultaneously deliver significant improvements in both accuracy and token efficiency on the challenging AIME24 benchmark. These results establish LASER as a promising reward design for enhancing the balance between efficacy and efficiency.

## 4 ADAPTIVE LENGTH-BASED STEP REWARD SHAPING

### 4.1 DESIGN PRINCIPLES

We highlight two key limitations not addressed in the design of LASER: (1) LASER requires a fixed target length, whereas the optimal length should be changed dynamically during training; (2) Additionally, different questions demand reasoning traces of varying lengths, with simple ones solvable by shorter reasoning and harder ones requiring longer deliberation.

Therefore, we extend LASER to be **D**ynamic and **D**ifficulty-aware, which we term as LASER-D. Rather than using a single fixed target length, our approach dynamically adjusts the target length throughout training and tailors it to questions of varying difficulty. Concretely, LASER-D *decouples the target length hyperparameter across different queries, allowing distinct target lengths to be assigned to various queries. Moreover, these target length hyperparameters are dynamically adjusted throughout training.*

We separate queries into three buckets of easy, medium, and hard difficulty levels, based on the correctness rates within the rollout batch – for each question, we have $k$ rollouts and use thresholds $k/3$ and $2k/3$ to separate them. As such, we have three distinct target length hyperparameters for these three query groups. Notably, we perform difficulty assessment for the queries during real-time RL training and use the training rollout batch, thus it only incurs negligible overhead on the computation. Being dynamic and difficulty-aware, one challenge raised is how to set the dynamic processes of the decoupled target length hyperparameters. Next, we introduce an *automatic* adapting mechanism, to adapt them without any manual intervention.

### 4.2 AUTOMATIC ADAPTING MECHANISM

LASER-D is driven by an automatic adapting mechanism that periodically evaluates and adjusts the target length parameters ($L_A$ in Table 3) for each difficulty level. Specifically, we first extract a small monitoring dataset $\mathcal{D}^M$ (e.g., 500 samples) from training data that mirrors the distribution of the training data. Every N training steps (e.g., 20), our approach searches and sets the target length hyperparameters based on this monitoring dataset.

Each query in $\mathcal{D}^M$ is assigned a difficulty label $d \in \{\text{easy}, \text{medium}, \text{hard}\}$ based on the number of correct rollouts among its $K$ sampled rollouts.[1]Since queries in different difficulty levels exhibit distinct length distributions, we compute the target length separately for each difficulty level. To do so, we introduce the Expected Correct Responses metric, $ECR_d$, which estimates how many complete, correct responses can be expected for queries of difficulty $d$ under a length limit $l$. Formally, for each candidate length $l$, we compute

$$ECR_d(l) = P_{l,d} \cdot C_d \tag{2}$$

where $P_{l,d}$ is the empirical coverage ratio, the fraction of all rollouts (computed over all monitoring queries with difficulty $d$) whose lengths do not exceed $l$. The constant $C_d$ is the minimum number of correct rollouts used to assign difficulty $d$ (e.g., $C_{\text{easy}}=6$, $C_{\text{medium}}=3$, $C_{\text{hard}}=1$ for $K=8$ here).

For each difficulty level $d$, the monitoring module enumerates candidate target lengths from the lower bound target length $L_T$ tokens up to the maximum context window (16,384 tokens) with an interval of $I$, computing coverage ratios $P_{l,d}$. We select the smallest target length $L_A$ satisfying $ECR_d \geq 1$ as the adaptive target length for each difficulty, ensuring at least one complete and correct response.

Intuitively, this mechanism sets the target length as the minimal generation length such that at least one rollout response is expected to be correct. This approach is reasonable because generating sequences shorter than this length would likely be detrimental, as correct responses are less probable. Conversely, generating longer sequences may be redundant, since correct responses can already be obtained with a shorter generation length.

**Selecting the Adaptive Target Length.** For each difficulty level $d$, we enumerate candidate lengths from the lower bound $L_T$ up to the context limit (16,384 tokens) with step size $I$. We select the smallest length $l$ satisfying $ECR_d(l) \geq 1$ as the adaptive target length $L_A$ for that difficulty.

---

[1]Practically, $K$ is set to be the same as the rollout size used during training, to maintain consistency with the training scenario.

Intuitively, this is the minimal length at which at least one correct response is expected; shorter limits harm correctness, while longer limits are redundant.

**Dynamic and Difficulty-Aware Reward**   During training, we apply these monitoring-derived parameters to implement dynamic and difficulty-aware rewards. Each training question's difficulty level is determined using the same classification method described earlier. Easier questions receive smaller target lengths $L_A$, while harder questions receive larger ones. Since monitoring runs every N steps, the difficulty-dependent target lengths are automatically adapted to the evolving policy model.

**Computational Efficiency**   This automatic adapting mechanism adds minimal computational overhead. By using a small monitoring dataset and evaluating only periodically, our method increases computation by just 3.5% in our experiments.

# 5   EXPERIMENTS

## 5.1   EXPERIMENTAL SETUP

**Setup**   We experiment with five capable and representative LRMs across four different sizes known for their overthinking tendencies: DeepSeek-R1-Distill-Qwen-1.5B, DeepSeek-R1-Distill-Qwen-7B, DeepSeek-R1-Distill-Qwen-32B, OpenReasoning-Nemotron-1.5B, DeepSeek-R1-Distill-Llama-8B. We adhere to the original prompt from DeepSeek-R1 (DeepSeek-AI et al., 2025), with the full prompt available in Appendix I.1. We train these models using the DeepScaleR-Preview-Dataset (Luo et al., 2025), a high-quality mathematics dataset containing 40K competition-level question-answer pairs. We leverage GRPO (Shao et al., 2024) as our online RL algorithm. We evaluate the models on four benchmarks of varying difficulty: MATH500 (Hendrycks et al., 2021b), OlympiadBench (He et al., 2024), AIME 2024, and AMC 2023. We set $\alpha = 0.5$ for our methods in all experiments to balance the trade-off between correctness rewards and solution length penalties. $L_T$ is a hyper-parameter for our approaches because the automatic adapting mechanism will enumerate the target length from $L_T$ to the context window size to select the adaptive target lengths $L_A$, as described in §4.2. Parameter settings for baseline methods are provided in Appendix I.3, and full details of our training procedure and evaluation methodology can be found in Appendix I.2.

**Baselines**   According to Table 3, we train models using different types of length rewards design and compare our LASER, LASER-D to previous works. Considering the high computational cost of RL training, we select Efficient Reason (Arora & Zanette, 2025) and L1-Max (Aggarwal & Welleck, 2025) as the representatives, since they perform better accuracy compared to other methods inside same group and are more close to our settings. For ThinkPrune (Hou et al., 2025), we re-evaluate their models. We also compare with the recent *full* mode method LAPO (Wu et al., 2025) and list two *adaptive* mode methods, Thinkless (Fang et al., 2025) and AutoThink (Tu et al., 2025) for reference.

## 5.2   EFFICACY-EFFICIENCY TRADE-OFF

Since there is a trade-off between accuracy and response length, one of the best ways to evaluate different methods is to compare the efficacy-efficiency trade-off. We start with the DeepSeek-R1-Distill-Qwen-1.5B model, as its small size allows us to run multiple experiments to investigate the trade-off of different approaches. To fully evaluate the potential of each method, we adjust key parameters ($\alpha$ for group-based reward, $L_T$ for other methods) to explore different tradeoffs along the accuracy-length trade-off curve. The full details of different hyper-parameters for different methods can be found in Table 13. As a result, each point in Figure 1/5 corresponds to a fully trained model under a distinct hyperparameter setting. Benchmark-level results are reported in Table 4, with additional truncation results in Appendix L.

As shown in Figure 1, LASER-D achieves the best efficacy–efficiency trade-off. On AIME2024 it reaches 34.2% accuracy with just 5,750 tokens (a 64% reduction), and across benchmarks (Figure 5) it obtains 60.3% accuracy with 3,520 tokens on average, compared to 10,177 for the original model. Compared to LASER, LASER-D benefits from its **dynamic** and **difficulty-aware** design, yielding consistent gains. LASER itself already outperforms prior baselines, while adaptive methods such as AutoThink and Thinkless collapse into extreme patterns, saving tokens on easy tasks (MATH) but remaining verbose on hard ones (AIME). Although these methods reduce tokens relative to their original models, but still consume around $10,000$ tokens on AIME. In contrast, both LASER and

Table 4: Accuracy (%) with average token usage for each dataset and different methods. Most important results in this table are visualized in Figure 1 and Figure 5 in Appendix B. The base model is DeepSeek-R1-Distill-Qwen-1.5B. "Original" denotes the original model. $T_k$ is the truncation method with context window $k$. L1 is *budget*-mode compression. Thinkless and AutoThink are based on *adaptive*. Other methods, including ours, are *full* mode. "Group" denotes the Efficient Reasoning (Arora & Zanette, 2025) with different $\alpha$. We only show the three most representative results here. For the full results, please refer to Tabel 14 in Appendix L.

| | Accuracy (%) | | | | | Generation Length (tokens) | | | | |
|---|---|---|---|---|---|---|---|---|---|---|
| | MATH 500 | AIME | AMC | Olympiad Bench | Avg. | MATH 500 | AIME | AMC | Olympiad Bench | Avg. |
| Original | 83.9 | 28.9 | 71.6 | 43.3 | 56.9 | 5042 | 15956 | 8202 | 11510 | 10177 |
| L1-Max-1024 | 76.4 | 15.0 | 59.4 | 39.1 | 47.5 | 661 | 1303 | 933 | 938 | 959 |
| L1-Max-4096 | 79.7 | 20.0 | 65.0 | 41.0 | 51.4 | 875 | 1718 | 1159 | 1229 | 1245 |
| Thinkless | 82.0 | 27.5 | 68.4 | 46.5 | 56.1 | 3702 | 10680 | 7244 | 7751 | 7344 |
| AutoThink | 84.0 | 34.6 | 67.0 | 44.8 | 57.6 | 2195 | 9514 | 5059 | 5559 | 5581 |
| $T_{8192}$ | 81.8 | 24.8 | 70.9 | 43.9 | 55.3 | 1795 | 4465 | 2560 | 2841 | 2915 |
| $T_{6144}$ | 80.9 | 20.2 | 66.2 | 42.1 | 52.3 | 1351 | 2821 | 1917 | 1947 | 2009 |
| $T_{4096}$ | 77.7 | 19.2 | 62.2 | 38.5 | 49.4 | 1054 | 2481 | 1484 | 1564 | 1646 |
| Group$_{\alpha=0.4}$ | 74.6 | 25.0 | 69.2 | 43.1 | 53.0 | 1069 | 4747 | 2162 | 2536 | 2629 |
| Group$_{\alpha=0.2}$ | 78.1 | 28.1 | 68.0 | 44.4 | 54.7 | 1135 | 5628 | 2635 | 2944 | 3085 |
| Group$_{\alpha=0.1}$ | 77.0 | 29.0 | 69.5 | 44.9 | 55.1 | 1228 | 6301 | 2808 | 3271 | 3402 |
| Group$_{\alpha=0.05}$ | 74.4 | 30.2 | 65.5 | 43.1 | 53.3 | 1193 | 4839 | 2457 | 2703 | 2798 |
| LAPO | 84.3 | 29.3 | 71.2 | 51.7 | 59.1 | 2354 | 8318 | 3568 | 4863 | 5581 |
| LASER$_{L_T=2048}$ | 83.6 | 29.2 | 71.6 | 44.1 | 57.1 | 1913 | 4815 | 2493 | 2767 | 2895 |
| LASER$_{L_T=4096}$ | 83.9 | 31.0 | 74.1 | 45.7 | 58.7 | 1914 | 5915 | 3136 | 3579 | 3636 |
| LASER$_{L_T=8192}$ | 85.6 | 31.5 | 75.9 | 47.7 | 60.2 | 2736 | 6589 | 4162 | 4547 | 4509 |
| LASER-D$_{L_T=1024}$ | 83.0 | 30.6 | 72.8 | 43.7 | 57.5 | 1362 | 4991 | 2560 | 2837 | 2862 |
| LASER-D$_{L_T=2048}$ | 82.2 | 31.0 | 73.3 | 46.2 | 58.2 | 1623 | 5158 | 2572 | 2960 | 3059 |
| LASER-D$_{L_T=4096}$ | 84.2 | 34.2 | 75.3 | 47.3 | 60.3 | 1872 | 5750 | 2981 | 3474 | 3520 |

LASER-D improve accuracy while reducing token usage across difficulties, advancing the frontier of *full*-mode methods.

**Ablation Studies** We conduct ablation studies focusing on two aspects: the effectiveness of the proposed **difficulty-aware** component and the **robustness** of our methods under different hyperparameter settings. We assess the necessity of the difficulty-aware mechanism by removing it from LASER-D. The results in Table 8 show that removing the difficulty-aware component leads to consistent accuracy drops across benchmarks, highlighting that dynamically adapting the target length to question difficulty is essential for achieving a better accuracy–efficiency trade-off. We examine the stability of our framework with respect to key hyperparameters, including the length reward coefficient $\alpha$ (Table 9), bucket thresholds (Table 10), and $ECR$ thresholds (Table 11). Across all these settings, our methods remain stable and maintain strong performance, confirming that our design is robust to hyperparameter choices. This also illustrates the robustness of our in-batch difficulty estimation strategy. Please refer to Appendix F.1 and Appendix F.2 for detailed results.

## 5.3 EXPERIMENTS ON LARGER MODELS AND MORE MODEL FAMILIES

We conduct experiments on larger models and different model families. Given the computational cost of larger models, we set key hyperparameters for each method to achieve an appropriate trade-off. Specifically, for DeepSeek-R1-Distill-Qwen-7B, we set $\alpha = 0.2$ for the group-based reward, $L_T = 8192$ for the truncation method in LASER, $L_T = 4096$ for LASER-D. Notably, we do not tune $\alpha$ with fixed value $0.5$ in all experiments of our methods. As shown in Table 5, LASER-D achieves the best trade-off with better accuracy and significantly fewer tokens. On the AIME dataset, it achieves an accuracy of $58.3\%$, representing a gain of $+\mathbf{5.2}$ points, while using only $5,379$ tokens—substantially fewer than the $13,414$ tokens used by the original model. Compared to other methods, LASER and LASER-D also attain better trade-offs on most benchmarks, particularly on the more challenging ones. For the 32B model, due to computational constraints, we compare the LASER-D-trained model with the original baseline under this larger setting and set $L_T = 8192$. LASER-D achieves comparable accuracy, while still significantly reducing output length. Notably, the accuracy of DeepSeek-R1-Distill-Qwen-32B on our training dataset is already very high—over $76\%$, leaving little room for further improvement. We speculate that with more challenging and diverse training data, LASER-D could yield further accuracy gains.

Beyond scaling within the DeepSeek-R1-Distill-Qwen family, we conducted experiments on three different LRMs: DeepSeek-R1-Distill-Llama-8B (based on LLaMA-3.1), OpenReasoning-Nemotron-1.5B (based on Qwen-2.5-Instruct) and DeepScaleR-1.5B. We apply LASER-D with $L_T = 4096$ for DeepSeek-R1-Distill-Llama-8B, DeepScaleR-1.5B and $L_T = 8192$ for OpenReasoning-Nemotron-

Table 5: Accuracy (%) with average token usage for each dataset and different methods using 7B and 32B models. "Original" denotes the original model. $T_k$ is the truncation method with context window $k$.

| | Accuracy (%) | | | | | Generation Length (tokens) | | | | |
|---|---|---|---|---|---|---|---|---|---|---|
| | MATH 500 | AIME | AMC | Olympiad Bench | Avg. | MATH 500 | AIME | AMC | Olympiad Bench | Avg. |
| DeepSeek-R1-Distill-Qwen-7B | | | | | | | | | | |
| Original | 92.6 | 53.1 | 88.4 | 58.9 | 73.3 | 4017 | 13414 | 6433 | 8987 | 8213 |
| $T_{8192}$ | 92.0 | 51.9 | 88.3 | 56.4 | 72.2 | 1972 | 5655 | 3159 | 3606 | 3598 |
| Group | 89.4 | 48.1 | 82.8 | 53.7 | 68.5 | 780 | 4271 | 1693 | 2348 | 2273 |
| LASER | 92.2 | 54.4 | 89.7 | 58.1 | 73.6 | 2317 | 6320 | 3733 | 4262 | 4158 |
| LASER-D | 92.2 | 58.3 | 90.0 | 61.0 | 75.4 | 1836 | 5379 | 2694 | 3350 | 3315 |
| DeepSeek-R1-Distill-Qwen-32B | | | | | | | | | | |
| Original | 94.4 | 71.7 | 93.1 | 64.6 | 80.9 | 3553 | 10335 | 6177 | 7697 | 6941 |
| LASER-D | 94.3 | 70.8 | 95.0 | 63.6 | 80.9 | 2210 | 6545 | 3569 | 4512 | 4209 |
| DeepSeek-R1-Distill-Llama-8B | | | | | | | | | | |
| Original | 89.4 | 49.2 | 90.0 | 55.4 | 71.0 | 4412 | 13760 | 6636 | 8970 | 8445 |
| LASER-D | 90.8 | 51.0 | 90.3 | 56.8 | 72.2 | 3937 | 8438 | 2768 | 5121 | 5075 |
| OpenReasoning-Nemotron-1.5B | | | | | | | | | | |
| Original | 84.5 | 47.7 | 83.9 | 55.5 | 67.9 | 8556 | 23486 | 13651 | 16493 | 15547 |
| LASER-D | 85.2 | 52.7 | 86.9 | 57.4 | 70.5 | 4506 | 14809 | 6670 | 9354 | 8835 |
| DeepScaleR-1.5B | | | | | | | | | | |
| Original | 87.3 | 40.3 | 82.3 | 50.8 | 65.1 | 3695 | 9550 | 5182 | 6628 | 6263 |
| LASER-D | 88.5 | 42.5 | 83.3 | 52.8 | 66.8 | 2100 | 5802 | 3978 | 3349 | 3807 |

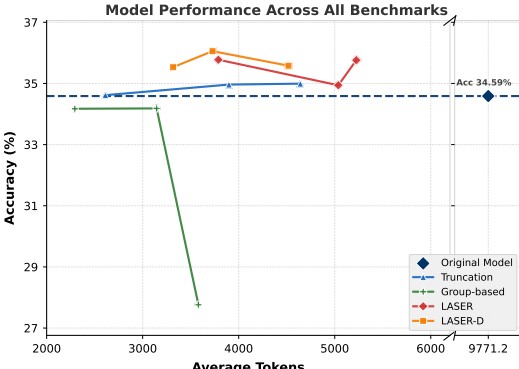

Figure 2: Performance on OOD benchmarks including GPQA (Rein et al., 2023), LSAT (Zhong et al., 2021; Wang et al., 2022), and MMLU (Hendrycks et al., 2021a). Please refer to Figure 6 for details on each benchmark.

1.5B, given its much longer original output length (20K+ tokens). Table 5 shows that LASER-D consistently delivers substantial accuracy improvements while reducing token usage by $\sim 50\%$, demonstrating strong generalizability.

## 5.4 EXPERIMENTS ON OUT-OF-DOMAIN BENCHMARKS

We evaluate whether LASER and LASER-D can generalize to domains outside the RL training distribution. We select three out-of-domain benchmarks: GPQA (Rein et al., 2023), LSAT (Wang et al., 2022), and MMLU (Hendrycks et al., 2021a), following the evaluation settings established by L1 (Aggarwal & Welleck, 2025), and LivecodeBench (Jain et al., 2025) for code generation. Figure 2 and Table 6 (full details in Appendix C) demonstrate that compared to the original model, LASER and LASER-D consistently achieve significant improvements in both accuracy and token usage, demonstrating robust generalization capabilities. And LASER-D maintain the best trade-off even compared to LASER.

## 6 ANALYSIS

We use DeepSeek-R1-Distill-Qwen-1.5B as the backbone, conduct a comprehensive analysis that includes budget-forcing inference (Appendix J), dynamics of adaptive target lengths (Appendix K), qualitative evaluations (Appendix N), and changes of thinking patterns.

Table 6: Accuracy (%) with average token usage on LiveCodeBench (Code Generation). The base model is DeepSeek-R1-Distill-Qwen-1.5B

|  | Accuracy (%) | Generation Length (tokens) |
|---|---|---|
| Original | 24.091 | 13120 |
| LASER-D$_{L_T=1024}$ | 27.955 | 6373 |
| LASER-D$_{L_T=2048}$ | 28.409 | 6420 |
| LASER-D$_{L_T=4096}$ | 28.409 | 6600 |

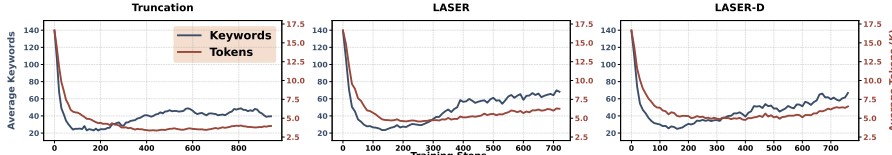

Figure 3: Average keyword amount and response length over RL training on AIME24. The truncation method uses a 8192 token context window. LASER and LASER-D employ a target length of $L_T = 2048$.

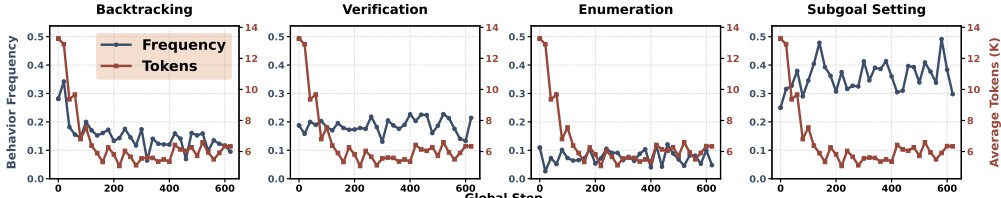

Figure 4: Changes in reasoning behaviors ratio and response length over RL training iterations on AIME2024. The figure shows how LASER-D's thinking patterns change during training with a target length of $L_T = 2048$.

## 6.1 CHANGES OF THINKING PATTERNS

To better understand the changes of response length, we analyze the changes of thinking patterns over RL iterations on AIME2024 with 16 samples per question. We analyze through two approaches, keywords counts (Yeo et al., 2025) and reasoning behavior ratios (Zeng et al., 2025).

**Shifts in "Self-Reflection" Keywords** "Self-reflection" or "Aha-moment" reasoning has emerged as an intriguing behavior in LRMs (DeepSeek-AI et al., 2025). Following previous work (Yeo et al., 2025), we track this behavior by monitoring seven representative keywords: [*"recheck"*,*"rethink"*,*"try again"*,*"wait"*, *"alternatively"*,*"retry"*,*"however"*]. Figure 3 shows that average keywords decline notably as response length decreases, suggesting RL reduces spurious "self-reflection". Interestingly, later training stages maintain shorter outputs while recovering moderate levels of "self-reflection", indicating more efficient use of this behavior.

**Changes in Thinking Behaviors** To more comprehensively capture shifts in thinking behavior, we employ proprietary large language models to conduct behavior-level analysis. We further analyze reasoning behaviors using the cognitive behavior framework of (Gandhi et al., 2025) with `gpt-4.1-mini`, focusing on four representative categories: *Backtracking*, *Verification*, *Subgoal Setting*, and *Enumeration*. Implementations and the full behavior list are provided in Appendix G. Figure 4 shows that *Backtracking* declines sharply (from $30\%$ to just above $10\%$) as response length is reduced, consistent with keyword statistics above, while the other behaviors remain stable, with a slight increase in *Subgoal Setting*. These results indicate shorter reasoning preserves core reasoning behaviors while minimizing unnecessary backtracking, rather than collapsing into shallow heuristics.

## 7 CONCLUSION

In this paper, we propose a unified view for RL-based CoT compression, unifying various reward-shaping and truncation methods. Building on this view, we introduce new approaches with adaptive, length-based reward shaping. Extensive experiments demonstrate that our proposed methods achieve superior efficacy-efficiency trade-off and significant improvements in both accuracy and token efficiency. Our analysis of reasoning behaviors reveals that our RL-based CoT compression effectively encourages models to reason more concisely and productively.

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

## A  THE USE OF LARGE LANGUAGE MODELS

For this paper, large language models(LLMs) are used solely for polishing the writing. The entire research process, including ideation, experimentation, and analysis—was conducted independently of LLM assistance.

## B  EFFICACY-EFFICIENCY TRADE-OFF

We illustrate the efficacy-efficiency trade-off in Figure 5. Our proposed methods, LASER and LASER-D, demonstrate significant improvements in both accuracy and token usage across all benchmarks, particularly in the most challenging ones. Notably, LASER-D achieves the best efficacy-efficiency trade-off compared to all other methods.

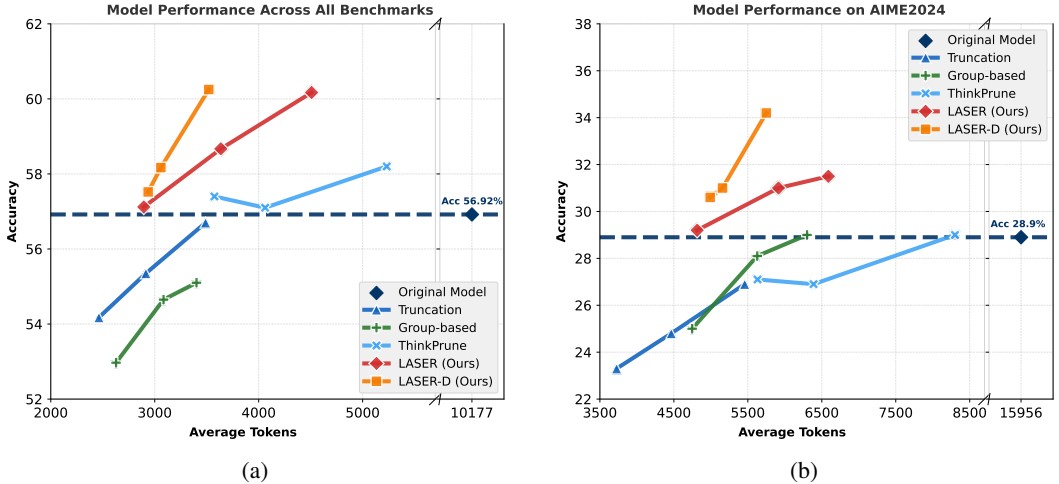

Figure 5: Trade-off between accuracy and response length across various methods. Each point represents a single training run with different hyper-parameters. Our methods, LASER-D and LASER, achieve the best trade-off compared to all other methods. (a) Accuracy and response length on all benchmarks (MATH500, AIME2024, AMC2023, Olympiad Bench) (b) Accuracy and response length on AIME2024

## C  EXPERIMENTS ON OUT-OF-DOMAIN BENCHMARKS

We evaluate whether LASER and LASER-D can generalize beyond their RL training distribution. Following the evaluation settings of L1 (Aggarwal & Welleck, 2025), we test on three out-of-domain benchmarks: GPQA (Rein et al., 2023), LSAT (Wang et al., 2022), and MMLU (Hendrycks et al., 2021a). Figure 6 shows the efficacy–efficiency trade-off on GPQA and the average performance across all benchmarks. Relative to the original model, both LASER and LASER-D achieve substantial improvements in accuracy and token efficiency, demonstrating strong generalization. Notably, LASER-D consistently maintains the best trade-off, outperforming LASER.

We further assess whether LASER-D can improve coding performance while reducing token usage. Using the LivecodeBench benchmark (Jain et al., 2025) under the OpenCompass framework (Contributors, 2023), we find that our method on DeepSeek-R1-Distill-Qwen-1.5B enhances reasoning accuracy while cutting token usage by roughly $50\%$ (Table 6).

## D  RATIO OF TRUNCATED RESPONSES DURING TRAINING WITH TRUNCATION

We analyze the ratio of truncated responses when applying an 8192 token limit during training. Our findings show that the proportion of truncated responses is initially very high—exceeding 45%, and remains substantial (above 10%) even after 200 rollout steps. This high truncation rate highlights the context window constraints in training is sub-optimal.

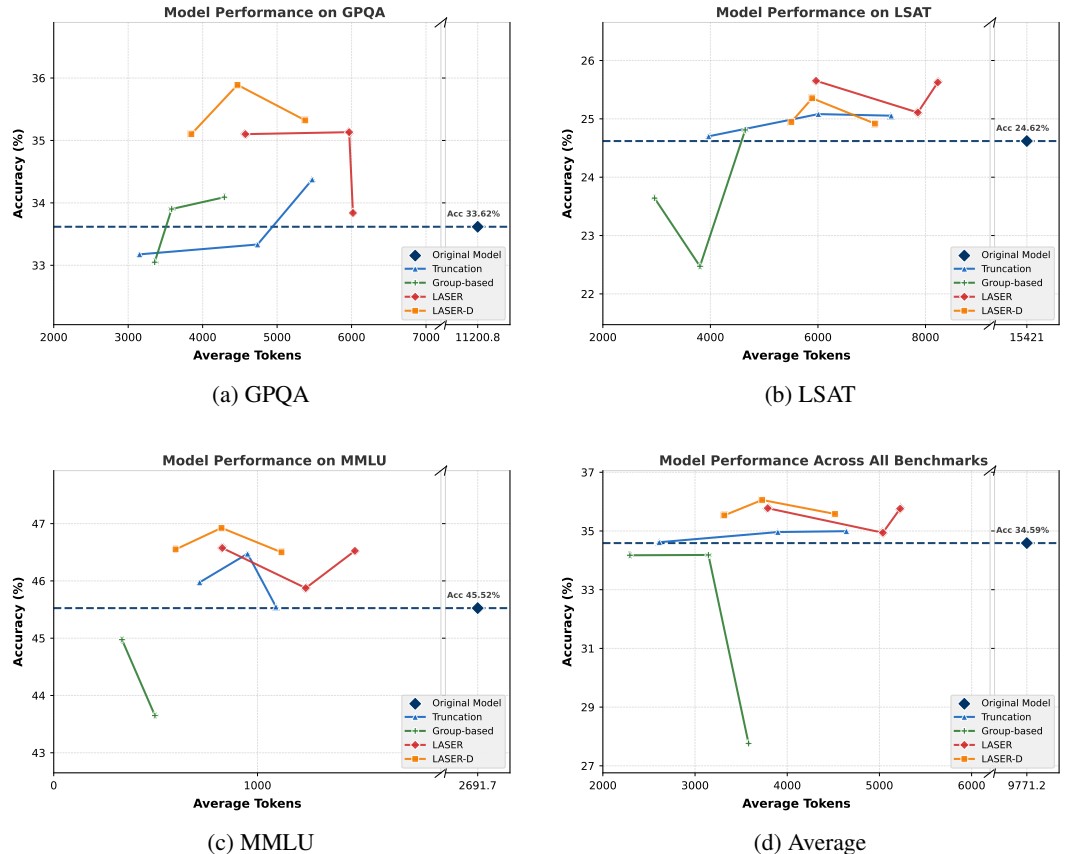

(a) GPQA

(b) LSAT

(c) MMLU

(d) Average

Figure 6: Performance on out-of-domain benchmarks including GPQA (Rein et al., 2023), LSAT (Zhong et al., 2021; Wang et al., 2022), and MMLU (Hendrycks et al., 2021a).

Table 7: Accuracy (%) with average token usage on LiveCodeBench (Code Generation). The base model is DeepSeek-R1-Distill-Qwen-1.5B

|  | Accuracy (%) | Generation Length (tokens) |
|---|---|---|
| Original | 24.091 | 13120 |
| LASER-D$_{L_T=1024}$ | 27.955 | 6373 |
| LASER-D$_{L_T=2048}$ | 28.409 | 6420 |
| LASER-D$_{L_T=4096}$ | 28.409 | 6600 |

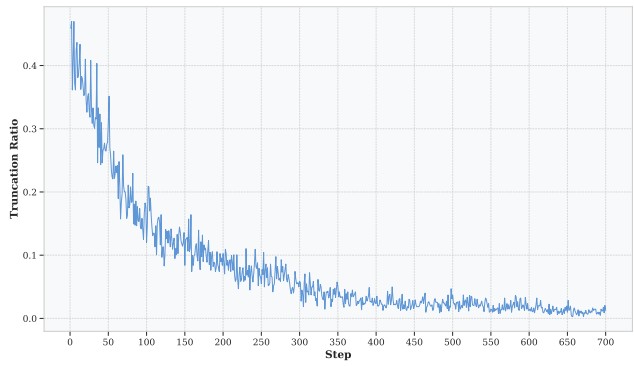

Figure 7: The ratio of truncated responses in training data with 8192 tokens limit.

# E    DYNAMICS OF ACCURACY AND REWARDS THROUGHOUT TRAINING

We present the accuracy and rewards for various methods across training iterations in Figure 8a and Figure 8b. As discussed in §3, group-based rewards tend to exploit the length rewards $S(y)$ while causing a significant drop in accuracy. Budget-based rewards such as L1-Max-16384 (Aggarwal & Welleck, 2025) suffer from unstable training when the context window is large. In contrast, other methods like truncation methods, LASER and LASER-Ddemonstrate a simultaneous increase in both rewards and accuracy throughout the training process.

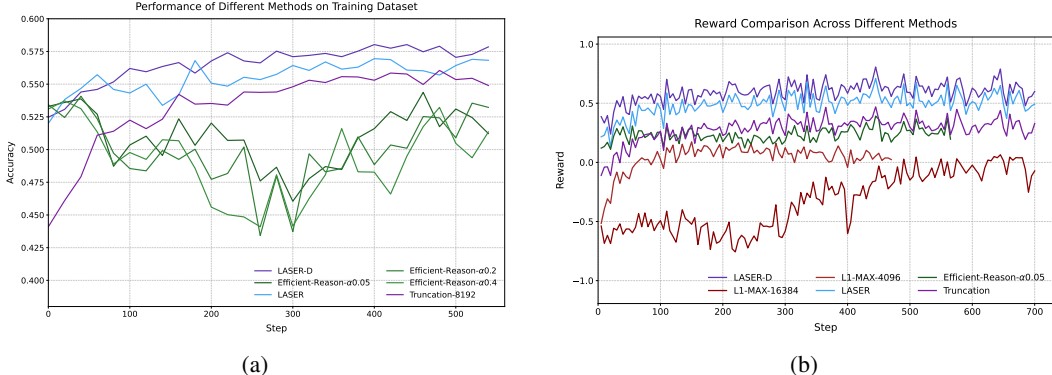

(a)    (b)

Figure 8: (a) Accuracy on training dataset across training iterations for different methods (b) Rewards across training iterations for different methods

# F    ABLATION STUDIES

## F.1    NECESSITY OF THE DIFFICULTY-AWARE COMPONENT

To evaluate the importance of the difficulty-aware component, we conducted an ablation study on LASER-D with $L_T = 2048$, where we removed the difficulty-aware mechanism and did not divide questions into the three difficulty buckets.

As shown in the Table 8, removing the difficulty-aware design leads to accuracy drops across most benchmarks. This result highlights that dynamically adapting the target length based on question difficulty is crucial for achieving a better accuracy–efficiency trade-off in our dynamic framework.

Table 8: Accuracy (%) with average token usage for each dataset and different methods.

| | Accuracy (%) | | | | | Generation Length (tokens) | | | | |
|---|---|---|---|---|---|---|---|---|---|---|
| | MATH 500 | AIME24 | AMC23 | Olympiad Bench | Avg. | MATH 500 | AIME24 | AMC23 | Olympiad Bench | Avg. |
| LASER-D$_{L_T=2048}$ | 82.2 | 31.0 | 73.3 | 46.2 | 58.2 | 1623 | 5158 | 2572 | 2960 | 3059 |
| w/o Difficulty-Aware | 82.3 | 29.0 | 70.9 | 45.8 | 57.0 | 1491 | 4693 | 2437 | 2767 | 2847 |

## F.2    ROBUSTNESS TO HYPER-PARAMETERS

To evaluate the stability and sensitivity of our methods, we conduct three groups of experiments on $\alpha$, bucket thresholds, and the n number for $ECR \geq n$.

**Different $\alpha$**    We conducted four experiments on LASER-Dusing different settings of the length reward coefficient $\alpha = 0.3, 0.7$ and target lengths $L_T = 2048, 4096$.

As shown in the Table 9, our method remains stable across different $\alpha$ values and consistently achieves the best trade-off between accuracy and token usage. Different $\alpha$ values have only a minor impact on the accuracy–token trade-off. Overall, $\alpha = 0.5$ is a strong choice, and other values also demonstrate sufficient robustness.

Table 9: Accuracy (%) with average token usage for each dataset and different $\alpha$ settings.

| | Accuracy (%) | | | | | Generation Length (tokens) | | | | |
|---|---|---|---|---|---|---|---|---|---|---|
| | MATH 500 | AIME24 | AMC23 | Olympiad Bench | Avg. | MATH 500 | AIME24 | AMC23 | Olympiad Bench | Avg. |
| LASER-D$_{L_T=2048,\alpha=0.5}$ | 82.2 | 31.0 | 73.3 | 46.2 | 58.2 | 1623 | 5158 | 2572 | 2960 | 3059 |
| LASER-D$_{L_T=4096,\alpha=0.5}$ | 84.2 | 34.2 | 75.3 | 47.3 | 60.3 | 1872 | 5750 | 2981 | 3474 | 3520 |
| LASER-D$_{L_T=2048,\alpha=0.3}$ | 82.7 | 31.7 | 74.2 | 44.9 | 58.4 | 1581 | 5272 | 2612 | 3049 | 3129 |
| LASER-D$_{L_T=2048,\alpha=0.3}$ | 84.7 | 33.9 | 75.0 | 46.8 | 60.1 | 2061 | 5835 | 3272 | 3611 | 3694 |
| LASER-D$_{L_T=2048,\alpha=0.7}$ | 82.1 | 30.4 | 72.7 | 45.0 | 57.6 | 1419 | 4982 | 2520 | 2812 | 2933 |
| LASER-D$_{L_T=4096,\alpha=0.7}$ | 83.3 | 33.2 | 75.5 | 47.8 | 60.0 | 1844 | 5303 | 2932 | 3318 | 3349 |

**Bucket Thresholds**   To assess the sensitivity of our method to bucket definitions, we conducted an ablation study using an alternative set of thresholds: defining questions answered correctly 1–3 times as *hard*, 4–6 times as *medium*, and 7–8 times as *easy*—compared to the original thresholds of 1–2, 3–5, and 6–8, respectively. This experiment was performed on LASER-D with a target length $L_T = 2048$. As shown in the Table 10, the new threshold configuration also leads to consistent improvements in both accuracy and efficiency. This result supports the robustness of our method and demonstrates that it is not overly sensitive to the specific bucket thresholds used for difficulty estimation.

Table 10: Accuracy (%) with average token usage under different difficulty settings.

| | Accuracy (%) | | | | | Generation Length (tokens) | | | | |
|---|---|---|---|---|---|---|---|---|---|---|
| | MATH 500 | AIME24 | AMC23 | Olympiad Bench | Avg. | MATH 500 | AIME24 | AMC23 | Olympiad Bench | Avg. |
| LASER-D$_{L_T=2048}$ (Original) | 82.2 | 31.0 | 73.3 | 46.2 | 58.2 | 1623 | 5158 | 2572 | 2960 | 3059 |
| 1–3 as hard, 4–6 as medium, 7–8 as easy | 82.7 | 30.0 | 73.1 | 47.1 | 58.2 | 1524 | 4934 | 2529 | 2825 | 2953 |

**Different $ECR$ thresholds**   To better illustrate the robustness of this setting, we conduct an ablation with $ECR \geq 2$ on LASER-D ($L_T = 2048$). As shown in the Table 11, the model still achieves improvements in both accuracy and efficiency compared to the original model. Compared to the $ECR \geq 1$ setting, the $ECR \geq 2$ version encourages the model to generate longer outputs—consistent with our design, since a stricter constraint requires the model to choose a larger $L_A$ to meet the criterion.

Overall, this experiment supports the stability of our method across different $ECR$ thresholds, and confirms that the design is flexible yet effective. And according to the experiment, $ECR \geq 1$ is already a reasonable choice for training.

Table 11: Accuracy (%) with average token usage under different ECR thresholds.

| | Accuracy (%) | | | | | Generation Length (tokens) | | | | |
|---|---|---|---|---|---|---|---|---|---|---|
| | MATH 500 | AIME24 | AMC23 | Olympiad Bench | Avg. | MATH 500 | AIME24 | AMC23 | Olympiad Bench | Avg. |
| LASER-D$_{L_T=2048,ECR\geq1}$ | 82.2 | 31.0 | 73.3 | 46.2 | 58.2 | 1623 | 5158 | 2572 | 2960 | 3059 |
| LASER-D$_{L_T=2048,ECR\geq2}$ | 82.5 | 31.0 | 73.0 | 46.5 | 58.2 | 1816 | 5694 | 3112 | 3365 | 3496 |

# G   ANALYSIS OF REASONING BEHAVIROS

We apply the cognitive behavior framework proposed by Gandhi et al. (2025) to conduct a detailed analysis of how reasoning behaviors change during our long-to-short RL. We use `gpt-4.1-mini` to perform a more fine-grained analysis of cognitive behaviors throughout the training process. Following Zeng et al. (2025), we use the prompt shown in Figure 11 to prompt `gpt-4.1-mini` to identify and analyze reasoning behaviors. We analyze these behaviors on AIME2024 by sampling one question 16 times, resulting in 480 responses for analysis. Since we start from a LRM, reasoning behaviors such as backtracking naturally appear in every response, especially for challenging benchmarks. We specifically track four key behaviors: *Backtracking*, *Verification*, *Enumeration*, and *Subgoal Setting*. For each behavior, we calculate its frequency ratio relative to all behaviors and

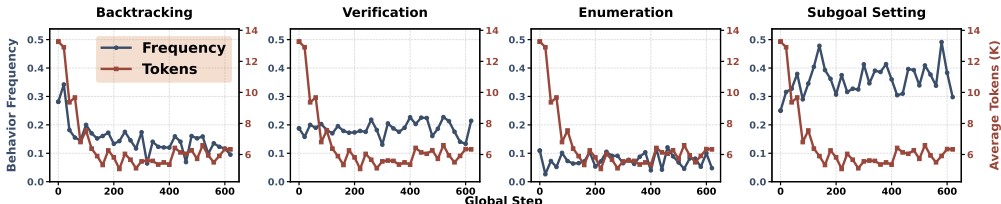

Figure 9: Average keyword amount and response length over RL training on AIME24. The truncation method uses a 8192 token context window. LASER and LASER-D employ a target length of $L_T = 2048$.

Figure 10: Changes in reasoning behaviors ratio and response length over RL training iterations on AIME2024. The figure shows how LASER-D's thinking patterns change during training with a target length of $L_T = 2048$.

report how these ratios change throughout the training process. The complete list of all reasoning behaviors analyzed is provided in Table 12.

Table 12: Complete list of reasoning behaviors

| **Reasoning Behavior** |
| --- |
| Subgoal Setting |
| Enumeration |
| Verification |
| Backtracking |
| Creative Analogy and Abstraction |
| Abstraction and Parametrization |
| Analytical Insight via Asymptotic Analysis |
| Creative Abstraction / Coordinate Setup |
| Use of Multiple Mathematical Tools and Identities |
| Creative Analogies and Insightful Generalizations |
| Algebraic Manipulation and Insightful Generalization |
| Abstraction to Modular Arithmetic and Divisibility |
| Creative Analogies and Abstractions |
| Insightful Generalization / Alternative Modeling |

# H  SUPPLEMENTARY DETAILS: LENGTH-BASED REWARD SHAPING FORMULATIONS

In this section, we provide additional details regarding the various formulations of length-based reward shaping as presented in Table 3. These formulations can viewed as different variants of our unified framework in Eq. 1 which can be implemented by making specific design choices for three key components: $C(y)$, $\lambda(y)$, and $S(y)$ inside the framework. Here we review the formulation of Eq. 1 to better illustrate following approaches.

$$\hat{R}(x, y) = C(y) + \lambda(y) \cdot S(y)$$

## H.1  TRUNCATION

**Vanilla Truncation**  As aforementioned discussions (§3), truncation is a special case of the length reward with $C(y) = 0$, where the target length $L_T$ is enforced by the context window. $\rho$ is set as 0. It follows the design:

> Prompt:
>
> Below is a chain-of-reasoning generated by a Language Model when attempting to solve a math problem. Evaluate this chain-of-reasoning to determine whether it demonstrates beneficial problem-solving behaviors that deviate from typical linear, monotonic reasoning patterns commonly observed in language models.
>
> <start_of_reasoning>
> {input}
> <end_of_reasoning>
>
> Specifically, actively identify and emphasize beneficial behaviors such as:
>
> (1) Backtracking: Explicitly revising approaches upon identifying errors or dead ends (e.g., "This approach won't work because...").
>
> (2) Verification: Systematically checking intermediate results or reasoning steps (e.g., "Let's verify this result by...").
>
> (3) Subgoal Setting: Breaking down complex problems into smaller, manageable steps (e.g., "To solve this, we first need to...").
>
> (4) Enumeration: Solving problems by exhaustively considering multiple cases or possibilities.
>
> Additionally, remain attentive to and encourage the identification of other beneficial behaviors not explicitly listed here, such as creative analogies, abstraction to simpler cases, or insightful generalizations.
>
> Important:
>
> Clearly specify each beneficial behavior you identify.
>
> Provide explicit examples from the reasoning chain.
>
> If no beneficial behaviors are observed, explicitly return an empty list.
>
> Provide your evaluation clearly, formatted as follows:
>
> ```json
> {
>   "behaviour": "",
>   "example": ""
> }
> ```

Figure 11: Prompt used to identify and analyze reasoning behaviors with `gpt-4.1-mini`

$$C(y) = 0$$
$$\lambda(y) = 1$$
$$S(y) = \begin{cases} R(x,y) & \text{if } L(y) \leq L_T \\ \rho & \text{if } L(y) > L_T \end{cases}$$

**ThinkPrune**   ThinkPrune (Hou et al., 2025) is another truncation-based approach, which extends vanilla truncation by introducing an adaptive target lengths $L_A$ to replace fixed target lengths $L_T$. $\rho$ is set as $0$. The design follows:

$$C(y) = 0$$
$$\lambda(y) = 1$$
$$S(y) = \begin{cases} R(x,y) & \text{if } L(y) \leq L_A \\ \rho & \text{if } L(y) > L_A \end{cases}$$

Their training methodology employs a progressive three-stage process with iterative refinement of $L_A$. Each subsequent stage initializes from the checkpoint of the previous stage while mannually reducing the value of $L_A$. Specifically, they progressively decrease $L_A$ through values of $4096$, $3072$, and $2048$ across the three stages.

## H.2   Group-based Rewards

In the context of group-based rewards, the length reward $S(y)$ is specifically designed to promote brevity by assigning higher scores to shorter responses within a rollout group. This mechanism functions as a comparison-based reward system that inherently favors more concise responses. Most of them follow the design $C(y) = R(x,y)$ to keep the accuracy performance of models.

**Efficient Reasoning**   Efficient Reasoning (Arora & Zanette, 2025) follows the principle of group-based reward by specifically encouraging conciseness within correct responses. The mean and variance scalars are computed exclusively from the subset of correct responses, ensuring appropriate statistical distributions. By selectively rewarding conciseness only when answers are correct, this approach maintains higher accuracy compared to Kimi-k1.5 (Kimi et al., 2025), which encourages wrong responses to be shorter. Considering the similarity between the two approaches and the better efficacy-efficiency trade-off, we select Efficient Reasoning as the representative group-based reward in this paper. The corresponding design can be formulated as follows:

$$C(y) = R(x,y)$$
$$\lambda(y) = \mathbb{I}(R)$$
$$S(y) = -\alpha \cdot \sigma \left( \frac{L(y) - Mean(y)}{STD(L)} \right)$$

**Kimi-k1.5**   The design of Kimi-k1.5 is similar to Efficient Reasoning (Arora & Zanette, 2025), with two main differences. First, the scalar factors are computed using the minimum response length and the difference between maximum response length and maximum length within a rollout group. Second, Kimi-k1.5 encourages all responses to be shorter, rather than focusing solely on shortening correct responses. Such a design has the potential to intensify reward hacking, as models may exploit the reward function by favoring shorter responses to maximize their scores. The designs follows:

$$C(y) = R(x,y)$$
$$\lambda(y) = 1$$
$$S(y) = \begin{cases} 0.5 - \frac{L(y) - L_{\min}}{L_{\max} - L_{\min}} & \text{if } \mathbb{I}(R) = 1 \\ \min\left(0, \ 0.5 - \frac{L(y) - L_{\min}}{L_{\max} - L_{\min}}\right) & \text{if } \mathbb{I}(R) = 0 \end{cases}$$

### H.3 BUDGET-BASED REWARD

Budget-based rewards use query-specific target lengths (budgets) and penalize responses that deviate from these instructions. And the coefficient $\alpha$ controls the trade-off between length reward and correctness reward. They come in two flavors: exact mode and max mode. We follow same settings as L1 (Aggarwal & Welleck, 2025) and set $\alpha = 0.0003$ for exact mode, $\alpha = 0.01$ for max mode.

**Exact Mode** In exact mode, the model must hit the specified target length $L_T$ exactly, and any deviation (even shorter outputs) is penalized. The design can be formulated as:

$$
\begin{aligned}
C(y) &= R(x, y) \\
\lambda(y) &= 1 \\
S(y) &= -\alpha \cdot |L(y) - L_T|
\end{aligned}
$$

**Max Mode** In max mode, only outputs that exceed $L_T$ incur a penalty. The designs follow:

$$
\begin{aligned}
C(y) &= 0 \\
\lambda(y) &= \mathbb{I}(R) \\
S(y) &= \mathrm{clip}(\alpha \cdot (L(y) - L_T) + \delta, 0, 1)
\end{aligned}
$$

## I TRAINING CONFIGURATIONS

We leverage the prompt from DeepSeek-AI et al. (2025), which is shown in Figure 12. And the mark for thinking is "<think>...</think>".

### I.1 TRAINING PROMPT

We list our training prompt in Figure 12, which follows the prompt from DeepSeek-R1 (DeepSeek-AI et al., 2025).

> Prompt:
>
> < | User | >{input}Please reason step by step, and put your final answer within \boxed{}.< | Assistant | >

Figure 12: Training prompt for our training.

### I.2 TRAINING AND EVALUATION DETAILS

We employ the verl (Sheng et al., 2024) framework for model training and Qwen-Math-Eval (Yang et al., 2024) for evaluation. During training, we set the rollout batch size to 128, conduct 8 rollouts per prompt, use a temperature of 0.6, and train with a mini-batch size of 64. In our preliminary experiments, we found long-to-short RL benefits from clip-higher strategy (Yu et al., 2025). So we follow DAPO (Yu et al., 2025) and set $\epsilon_{high}$ as 0.28. For evaluation, we maintain a sampling temperature of 0.6 and permit a maximum of 32,768 tokens to be generated. The number of samplings during evaluation is contingent on the dataset size: 4 samples per question for MATH500 and OlympiadBench, and 16 samples for AIME 2024 and AMC 2023.

### I.3 FULL HYPER-PARAMETER LIST FOR DIFFERENT LENGTH-BASED REWARDS

We list the all hyper-parameters for $L_T$, $\alpha$ and $L_A$ in Table 13.

## J BUDGET-FORCING INFERENCE

To further analyze the impact of different length rewards, we conduct experiments using the budget-forcing setup introduced in S1 (Muennighoff et al., 2025), which restricts the model to stop reasoning

Table 13: The details of key hyper-parameters for different methods

| Methods | Hyper-Parameters |
|---|---|
| Truncation | $L_T = [10240, 8192, 7168, 6144, 4098, 2048]$ |
| Think-Prune | $L_A = [4096, 3072, 2048]$ |
| Group-Based Rewards | $\alpha = [0.4, 0.2, 0.1, 0.05]$ |
| L1-Max | $\alpha = 0.002$ |
| LASER | $L_T = [8192, 4096, 2048]$ |
| LASER-D | $L_T = [4096, 2048, 1024]$ |

after a fixed number of tokens $B$. We adopt their experimental setting and evaluate across $B = [500, 1000, 2000, 4000, 8000]$ We follow the budget-forcing implementations of Muennighoff et al. (2025); Hou et al. (2025). Specifically, we follow their implementations and modify the codebase of Qwen-Math-Eval. We stop the thinking process of LRMs by appending "</think>\n\n**Final Answer.**". Since empirically, DeepSeek-R1-Distill-Qwen-1.5B typically summarize its final answer starting with "\n\n**Final Answer.**". We use the same settings as our evaluations where we sample responses for multiple times with $\text{temperature} = 0.6$.. As shown in Figures 13a and 13b, despite not being explicitly trained with any budget-forcing mechanisms, LASER-D consistently achieve strong trade-offs between accuracy and token efficiency, particularly on harder questions or when inference budgets are moderately constrained.

While LASER performs competitively on average benchmarks, it lags behind LASER-D/LASER-DE under strict token budgets or on more challenging examples. L1-Max, specifically trained to meet varying budget constraints during training, performs best under extremely tight budgets, demonstrating the strength of budget-specific optimization. However, its performance plateaus when more budget is available, limiting its ability to improve on harder tasks and resulting in a suboptimal trade-off, as shown in Figure 13b. Group-based methods are also effective in low-budget scenarios due to their reward structure favoring shorter outputs, though this often leads to overly brief responses. ThinkPrune shows comparable performance to LASER under looser budgets but inherits the limitations of truncation-based approaches, struggling on difficult problems even when more tokens are available.

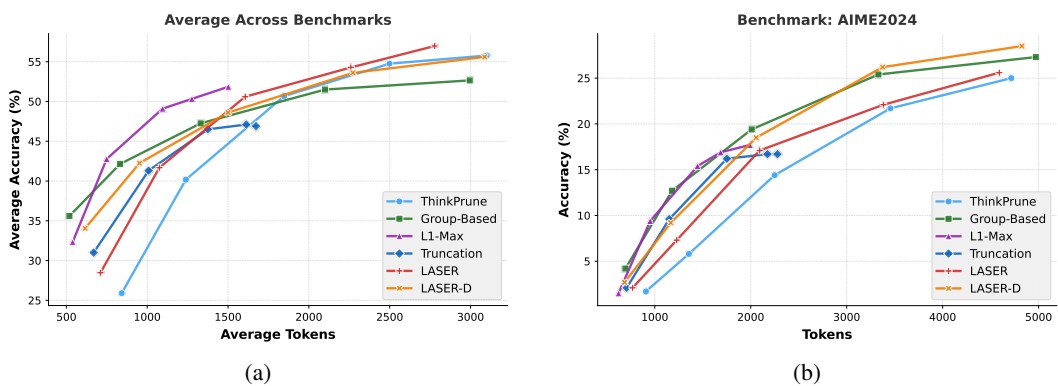

(a)                                             (b)

Figure 13: Budget-forcing inference with different methods. (a) Average accuracy with different output budget on all benchmarks (b) The accuracy of different methods on AIME2024 with different output budget.

## K   DYNAMICS OF ADAPTIVE TARGET LENGTHS

In this section, we analyze the dynamics of adaptive target lengths during the training process of LASER-D. Figure 14 shows how the adaptive target length $L_A$ changes over training iterations for both methods.

As demonstrated in Figure 14, our method dynamically selects appropriate target lengths based on problem difficulty. For easy problems (left figure), the model quickly identifies that shorter target

lengths are sufficient. For medium-difficulty problems (middle figure), the model begins with longer target lengths (10,000+) and gradually reduces them to 3000-4000 as training continues. For difficult problems (right figure), the model consistently maintains target lengths near the maximum context window size, with some fluctuations attributable to computational precision issues. This adaptive behavior highlights the effectiveness of our approach in efficiently allocating computational resources based on problem complexity.

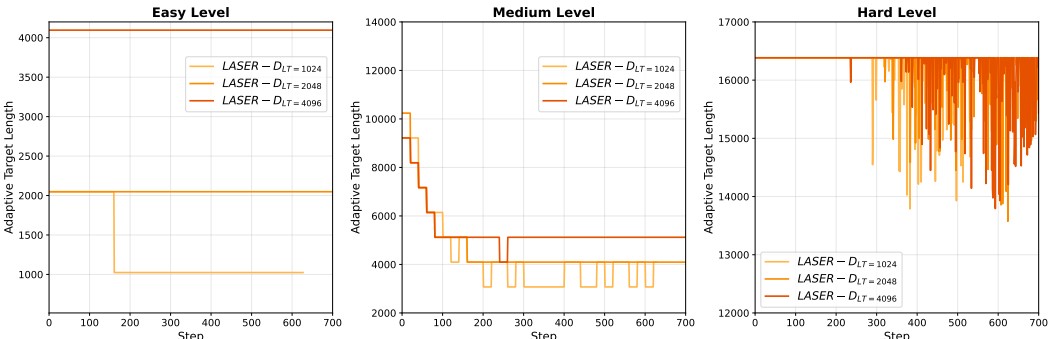

Figure 14: Dynamics of adaptive target lengths during the training process of LASER-D. The figure shows how the adaptive target length $L_A$ changes over training iterations for problems of different difficulty levels (easy, medium, hard). For easy problems, the model selects short target lengths; for medium problems, it gradually decreases from higher initial values; and for hard problems, it maintains consistently high target lengths near the context window limit. This demonstrates the methods' ability to adaptively allocate computational resources based on problem complexity, unlike fixed-length approaches.

## L    FULL MAIN RESULTS

We list the full results of different methods in Table 14.

## M    VISUALIZATION DETAILS

In this appendix, we provide details about the visualization of different reward functions depicted in Table 3. These visualizations illustrate how different methods calculate rewards based on response length.

### M.1    VISUALIZATION PARAMETERS

Each visualization captures the relationship between response length and reward value with the following specifications:

- **X-axis**: L(y) represents the response length, ranging from 0 to 20 tokens.
- **Y-axis**: Reward value, with different ranges depending on the method.
- **Line styles**: Solid lines represent rewards for correct responses (blue), while dashed lines represent rewards for incorrect responses (red).
- **Target length** ($L_T$): Set to 10 tokens for all methods.

The visualizations were generated using a high-resolution grid of 400 points between 0 and 20 tokens.

### M.2    UNIFIED REWARD FORMULATION

Each method can be represented using the unified reward formula:

$$\hat{R}(x, y) = C(y) + \lambda(y) \cdot S(y)$$

We implement the specific components for each method in this simulation as follows. Note that the paramters are only used for better visualization which are different from the practical experiments.

Table 14: Full results of accuracy (%) with average token usage for each dataset and different methods. Most important results in this table are visualized in Figure 1 and Figure 5 in Appendix B. The base model is DeepSeek-R1-Distill-Qwen-1.5B. "Original" denotes the original model. $T_k$ is the truncation method with context window $k$. L1 is *budget*-mode compression. Thinkless and AutoThink are based on *adaptive*. Other methods, including ours, are *full* mode. "Group" denotes the Efficient Reasoning (Arora & Zanette, 2025) with different $\alpha$.

| | Accuracy (%) | | | | | Generation Length (tokens) | | | | |
|---|---|---|---|---|---|---|---|---|---|---|
| | MATH 500 | AIME | AMC | Olympiad Bench | Avg. | MATH 500 | AIME | AMC | Olympiad Bench | Avg. |
| Original | 83.9 | 28.9 | 71.6 | 43.3 | 56.9 | 5042 | 15956 | 8202 | 11510 | 10177 |
| L1-Max-1024 | 76.4 | 15.0 | 59.4 | 39.1 | 47.5 | 661 | 1303 | 933 | 938 | 959 |
| L1-Max-4096 | 79.7 | 20.0 | 65.0 | 41.0 | 51.4 | 875 | 1718 | 1159 | 1229 | 1245 |
| Thinkless | 82.0 | 27.5 | 68.4 | 46.5 | 56.1 | 3702 | 10680 | 7244 | 7751 | 7344 |
| AutoThink | 84.0 | 34.6 | 67.0 | 44.8 | 57.6 | 2195 | 9514 | 5059 | 5559 | 5581 |
| $T_{10240}$ | 82.7 | 26.9 | 73.1 | 44.1 | 56.7 | 2056 | 5458 | 3036 | 3405 | 3489 |
| $T_{8192}$ | 81.8 | 24.8 | 70.9 | 43.9 | 55.35 | 1795 | 4465 | 2560 | 2841 | 2915 |
| $T_{7168}$ | 81.8 | 23.3 | 68.6 | 43.0 | 54.18 | 1553 | 3726 | 2251 | 2323 | 2463 |
| $T_{6144}$ | 80.9 | 20.2 | 66.2 | 42.1 | 52.35 | 1351 | 2821 | 1917 | 1947 | 2009 |
| $T_{4096}$ | 77.7 | 19.2 | 62.2 | 38.5 | 49.4 | 1054 | 2481 | 1484 | 1564 | 1646 |
| $T_{2048}$ | 73.2 | 15.8 | 56.9 | 35.9 | 45.45 | 721 | 1029 | 936 | 1084 | 943 |
| Group$_{\alpha=0.4}$ | 74.6 | 25.0 | 69.2 | 43.1 | 53.0 | 1069 | 4747 | 2162 | 2536 | 2629 |
| Group$_{\alpha=0.2}$ | 78.1 | 28.1 | 68.0 | 44.4 | 54.7 | 1135 | 5628 | 2635 | 2944 | 3085 |
| Group$_{\alpha=0.1}$ | 77.0 | 29.0 | 69.5 | 44.9 | 55.1 | 1228 | 6301 | 2808 | 3271 | 3402 |
| Group$_{\alpha=0.05}$ | 74.4 | 30.2 | 65.5 | 43.1 | 53.3 | 1193 | 4839 | 2457 | 2703 | 2798 |
| LAPO | 84.3 | 29.3 | 71.2 | 51.7 | 59.1 | 2354 | 8318 | 3568 | 4863 | 5581 |
| LASER$_{L_T=2048}$ | 83.6 | 29.2 | 71.6 | 44.1 | 57.1 | 1913 | 4815 | 2493 | 2767 | 2895 |
| LASER$_{L_T=4096}$ | 83.9 | 31.0 | 74.1 | 45.7 | 58.7 | 1914 | 5915 | 3136 | 3579 | 3636 |
| LASER$_{L_T=8192}$ | 85.6 | 31.5 | 75.9 | 47.7 | 60.2 | 2736 | 6589 | 4162 | 4547 | 4509 |
| LASER-D$_{L_T=1024}$ | 83.0 | 30.6 | 72.8 | 43.7 | 57.5 | 1362 | 4991 | 2560 | 2837 | 2862 |
| LASER-D$_{L_T=2048}$ | 82.2 | 31.0 | 73.3 | 46.2 | 58.2 | 1623 | 5158 | 2572 | 2960 | 3059 |
| LASER-D$_{L_T=4096}$ | 84.2 | 34.2 | 75.3 | 47.3 | 60.3 | 1872 | 5750 | 2981 | 3474 | 3520 |

**Vanilla Truncation**

$$C(y) = 0$$
$$\lambda(y) = 1$$
$$S(y) = \begin{cases} R(x,y) & \text{if } L(y) \leq L_T \\ \rho & \text{if } L(y) > L_T \end{cases}$$

where $L_T = 10$ and $\rho = 0$.

**ThinkPrune**

$$C(y) = 0$$
$$\lambda(y) = 1$$
$$S(y) = \begin{cases} R(x,y) & \text{if } L(y) \leq L_A \\ \rho & \text{if } L(y) > L_A \end{cases}$$

where $L_A \in \{10, 7.5, 5\}$.

**Efficient Reasoning**

$$C(y) = R(x,y)$$
$$\lambda(y) = \mathbb{I}(R)$$
$$S(y) = -\alpha \cdot \sigma \left( \frac{L(y) - Mean(y)}{STD(L)} \right)$$

where $\mu = 10$ and $\sigma = 2$.

**Kimi-k1.5**

$$C(y) = R(x, y)$$
$$\lambda(y) = 1$$
$$S(y) = \begin{cases} 0.5 - \frac{L(y) - L_{\min}}{L_{\max} - L_{\min}} & \text{if } \mathbb{I}(R) = 1 \\ \min\left(0, \ 0.5 - \frac{L(y) - L_{\min}}{L_{\max} - L_{\min}}\right) & \text{if } \mathbb{I}(R) = 0 \end{cases}$$

where $L_{min} = 2.5$ and $L_{max} = 20$.

**L1-Exact**

$$C(y) = R(x, y)$$
$$\lambda(y) = 1$$
$$S(y) = -\alpha \cdot |L(y) - L_T|$$

where $\alpha = 0.03$ and $L_T = 10$.

**L1-Max**

$$C(y) = 0$$
$$\lambda(y) = \mathbb{I}(R)$$
$$S(y) = \text{clip}(\alpha \cdot (L(y) - L_T) + \delta, 0, 1)$$

where $\alpha = 0.03$ and $L_T = 10$.

**LASER**

$$C(y) = R(x, y)$$
$$\lambda(y) = \mathbb{I}(R)$$
$$S(y) = \alpha \cdot \mathbb{I}(L(y) < L_T)$$

where $L_T = 10$.

**LASER-D**

$$C(y) = R(x, y)$$
$$\lambda(y) = \mathbb{I}(R)$$
$$S(y) = \alpha \cdot \mathbb{I}(L(y) < L_A)$$

where $L_A \in \{10, 7.5, 5\}$.

For methods with multiple adaptive target lengths $L_A$ values (ThinkPrune, LASER-D, and LASER-DE), different shades of the base colors were used:

- Correct responses (blue): RGB(26,71,142), RGB(62,101,184), RGB(125,154,230)
- Incorrect responses (red): RGB(139,0,0), RGB(183,50,40), RGB(224,93,86)

## N    QUALITATIVE ANALYSIS ON EFFICIENT REASONING

We conduct a qualitative analysis on the trivial question "1+1=?" and the MATH500 dataset to understand how RL improves reasoning efficiency. Comparing the original DeepSeek-R1-Distill-Qwen-1.5B model with the LASER-D-trained version, Figure 1 illustrates how the original model generates repetitive "self-reflection" even for trivial questions, while the trained model directly provides the answer. Our analysis of MATH500 (detailed in Figure 15, Figure 16 and Figure 17) reveals that the original model tends towards verbose, redundant explanations of single ideas. In contrast, the LASER-D-trained model expresses the same concepts more succinctly using structured formulas, significantly improving token efficiency. This suggests our RL-based approach not only reduces unproductive backtracking but also encourages a shift towards more concise and direct expression.

Question: If $f(x) = \frac{3x-2}{x-2}$, what is the value of $f(-2) +f(-1)+f(0)$? Express your answer as a common fraction.

<think>
Okay, so I need to find the distance between two points: (2, -6) and (-4, 3). Hmm, I remember from school that there's a formula for this kind of problem. Let me think. Is it something with the distance formula? Yeah, I think it's called the distance formula. It's used to find the distance between two points in a coordinate plane.

Let me try to recall how that formula works. I think it's something like taking the difference in the x-coordinates, squaring it, taking the difference in the y-coordinates, squaring that, adding them together, and then taking the square root. Yeah, that sounds right. So, if I have two points, let's say (x1, y1) and (x2, y2), the distance d between them is calculated by the formula:

d = sqrt[(x2 - x1)^2 + (y2 - y1)^2]

Alright, let me write that down step by step so I don't get confused.

First, I need to identify the coordinates of both points. The first point is (2, -6), so that means x1 is 2 and y1 is -6. The second point is (-4, 3), so x2 is -4 and y2 is 3.

Now, I need to find the differences in the x and y coordinates. So, subtract x1 from x2, which is (-4 - 2). Let me calculate that. Hmm, -4 minus 2 is -6. So, that part is -6. Next, subtract y1 from y2, which is (3 - (-6)). Wait, subtracting a negative is like adding the positive, so that would be 3 + 6, which is 9.

So now, I have the differences: -6 for the x-coordinate and 9 for the y-coordinate. But since we're going to square them, the negative sign doesn't matter because squaring a negative number makes it positive. So, (-6)^2 is 36, and 9^2 is 81.

Now, I need to add those squared differences together. 36 plus 81 is... let me add that. 36 plus 80 is 116, and then plus 1 more is 117. So, the sum inside the square root is 117.

Therefore, the distance d is the square root of 117. Now, I need to simplify that radical. Let me see if 117 has any square factors. Hmm, 117 divided by 9 is 13. So, 9 is a perfect square, and 13 is a prime number. So, I can write sqrt(117) as sqrt(9 * 13). Since sqrt(9) is 3, that becomes 3*sqrt(13).

Wait, let me make sure I did that correctly. So, 9 times 13 is 117, and sqrt(9) is 3, so yes, that would make sense. So, putting it together, the distance is 3 times the square root of 13.

Let me just verify the steps again to make sure I didn't make a mistake. Coordinates: (2, -6) and (-4, 3). Compute x2 - x1: -4 - 2 = -6. y2 - y1: 3 - (-6) = 9. Square them: (-6)^2 = 36; 9^2 = 81. Sum: 36 + 81 = 117. Square root: sqrt(117). Factor: 117 = 9 * 13, so sqrt(9*13) = 3*sqrt(13). Yeah, that all looks correct.

Is there another way to approach this problem? Maybe by plotting the points on a graph and using the Pythagorean theorem? Let me think. If I plot (2, -6) and (-4, 3), the distance between them would be the hypotenuse of a right-angled triangle where the other two sides are the differences in x and y coordinates. So, the horizontal leg would be the distance between x = 2 and x = -4, which is indeed 6 units. The vertical leg would be the distance between y = -6 and y = 3, which is 9 units. Then, using Pythagoras, the hypotenuse would be sqrt(6^2 + 9^2) = sqrt(36 + 81) = sqrt(117). So, same result as before, which is reassuring.

Another thought: Could I have mixed up the order of subtraction? Like, if I had done x1 - x2 instead of x2 - x1, would that change anything? Let me see. If I do x1 - x2, that would be 2 - (-4) = 2 + 4 = 6, same as before. And y1 - y2 is -6 - 3 = -9, which when squared is also 81. So, same result: 6^2 + (-9)^2 = 36 +81 = 117. So, that's consistent.

...

[2145 Tokens]

<think>
Okay, so I need to find the distance between the two points (2, -6) and (-4, 3). Hmm, I remember there's a formula for the distance between two points in a coordinate plane. It's something like the square root of the sum of the squares of the differences in each coordinate. Let me write that down.

The formula is d = sqrt[(x2 - x1)^2 + (y2 - y1)^2]. So I can plug in the coordinates into this formula.

Let me assign (x1, y1) as (2, -6) and (x2, y2) as (-4, 3). Then the differences would be x2 - x1 = -4 - 2 = -6 and y2 - y1 = 3 - (-6) = 9.

So now I square those differences: (-6)^2 = 36 and 9^2 = 81. Then I add those together: 36 + 81 = 117.

Finally, the distance is the square root of 117. I should check if this can be simplified. The factors of 117 are 9*13, so sqrt(9*13) = 3*sqrt(13).

So the distance is 3Ã13 units.

**Final Answer**
The distance between the points is \boxed{3\sqrt{13}} units.
</think>

To find the distance between the points \((2, -6)\) and \((-4, 3)\), we use the distance formula:

\[
d = \sqrt{(x_2 - x_1)^2 + (y_2 - y_1)^2}
\]

Substituting the given points \((2, -6)\) as \((x_1, y_1)\) and \((-4, 3)\) as \((x_2, y_2)\), we calculate the differences:

\[
x_2 - x_1 = -4 - 2 = -6
\]
\[
y_2 - y_1 = 3 - (-6) = 9
\]

Squaring these differences and adding them:

\[
(-6)^2 + 9^2 = 36 + 81 = 117
\]

Taking the square root:

\[
\sqrt{117} = 3\sqrt{13}
\]

Thus, the distance between the points is \(\boxed{3\sqrt{13}}\) units.

[550 Tokens]

Figure 15: The full example of Figure 1

Question: Define $p = \sum_{k = 1}^\infty \frac{1}{k^2} \quad \text{and} \quad q = \sum_{k = 1}^\infty \frac{1}{k^3}.$ Find a way to write $\sum_{j = 1}^\infty \sum_{k = 1}^\infty \frac{1}{(j + k)^3}$ in terms of $p$ and $q.$

<think>
Okay, so I have this problem where I need to express a double sum involving reciprocals of cubes as a combination of two known sums, p and q. Let me write down what I know.

First, p is defined as the sum from k = 1 to infinity of $1/k^2$. I remember that p is known as the Basel problem, and its value is $\pi^2/6$. But I don't know if I need the exact value of p, or if I can just keep it as p for now.

Similarly, q is the sum from k = 1 to infinity of $1/k^3$, and I think that's called the Apery's constant. I don't recall its exact value, but again, maybe I don't need to compute it exactly.

The main goal is to find an expression for the double sum:

$S = \sum_{j=1}^\infty \sum_{k=1}^\infty 1/(j + k)^3$.

And express this S in terms of p and q.

Hmm, okay. So, I need to work with S, which is a double sum over j and k starting at 1, and each term is 1 over (j + k) cubed.

I remember that sometimes, double sums can be transformed into single sums by changing variables or using some symmetry. Maybe I can find a substitution that simplifies this.

Let me think: in the double sum, the indices j and k are symmetric. So perhaps I can make a substitution where I set n = j + k, and then see how many times each n appears in the double sum.

That is, for each n ≥ 2 (since j and k start at 1), the number of pairs (j, k) such that j + k = n is (n - 1). For example, when n=2, only (1,1); when n=3, (1,2) and (2,1); for n=4, (1,3), (2,2), (3,1), etc.

Therefore, I can rewrite the double sum S as:

$S = \sum_{n=2}^\infty (n - 1) * [1/n^3]$.

Because for each n, there are (n - 1) pairs (j, k) where j + k = n, and each of these contributes $1/n^3$ to the sum.

So now, S can be written as:

$S = \sum_{n=2}^\infty (n - 1)/n^3$.

Let me simplify this expression. Let's split the fraction:

$(n - 1)/n^3 = n/n^3 - 1/n^3 = 1/n^2 - 1/n^3$.

Therefore, $S = \sum_{n=2}^\infty (1/n^2 - 1/n^3)$.

Now, let's write this as:

$S = \sum_{n=2}^\infty 1/n^2 - \sum_{n=2}^\infty 1/n^3$.

But wait, the original p is $\sum_{n=1}^\infty 1/n^2$, which is 1 + $\sum_{n=2}^\infty 1/n^2$. Similarly, q is $\sum_{n=1}^\infty 1/n^3$, which is 1 + $\sum_{n=2}^\infty 1/n^3$.

So, if I subtract 1 from p, I get $\sum_{n=2}^\infty 1/n^2 = p - 1$.

Similarly, subtracting 1 from q gives $\sum_{n=2}^\infty 1/n^3 = q - 1$.

Therefore, substituting back into S:

$S = (p - 1) - (q - 1) = p - 1 - q + 1 = p - q$.

Wait, but that can't be right. Because when I split the sum $S = \sum_{n=2}^\infty (1/n^2 - 1/n^3) = \sum_{n=2}^\infty 1/n^2 - \sum_{n=2}^\infty 1/n^3 = (p - 1) - (q - 1) = p - q$. So, is S = p - q?

But hold on, let me check that. Let me compute S numerically for a few terms to see if this holds.

...
Thus, substituting these into the expression for $S$, we get:
$S = (p - 1) - (q - 1) = p - q.$

Therefore, the final answer is:
$\boxed{p - q}$

[Total Tokens: 17523]

<think>
Okay, so I have this problem where I need to express a double sum in terms of p and q, which are known as the sums of reciprocals of squares and cubes respectively. Let me write down what I need to find:

The sum is over all j and k from 1 to infinity of $1/(j + k)^3$. So,

$S = \sum_{j=1}^\infty \sum_{k=1}^\infty 1/(j + k)^3$.

And $p = \sum_{k=1}^\infty 1/k^2$, $q = \sum_{k=1}^\infty 1/k^3$.

...

Therefore,

$S = \sum_{m=2}^\infty (m - 1)/m^3$.

So $S = \sum_{m=2}^\infty (m - 1)/m^3 = \sum_{m=2}^\infty (1/m^2 - 1/m^3)$.

Which can be written as:

$S = \sum_{m=2}^\infty 1/m^2 - \sum_{m=2}^\infty 1/m^3$.

We know $p = \sum_{m=1}^\infty 1/m^2$, so $\sum_{m=2}^\infty 1/m^2 = p - 1$.

Similarly, $q = \sum_{m=1}^\infty 1/m^3$, so $\sum_{m=2}^\infty 1/m^3 = q - 1$.

Therefore, $S = (p - 1) - (q - 1) = p - q$.

...

Thus, the expression becomes:

$(p - 1) - (q - 1) = p - q$

The final answer is:

$\boxed{p - q}$

[Total Tokens: 1114]

Figure 16: Additional case study demonstrating the evolution of reasoning efficiency. In this example, the original model required over 17K tokens to solve a question from the MATH500 dataset, while our trained model accomplished the same task using only 1K+ tokens.

Question: If $f(x) = \frac{3x-2}{x-2}$, what is the value of $f(-2) + f(-1) + f(0)$? Express your answer as a common fraction.

Okay, so I have this function f(x) which is equal to (3x - 2) divided by (x - 2). The question is asking me to find the value of f(-2) plus f(-1) plus f(0) and express the answer as a common fraction. Hmm, let me break this down step by step because I want to make sure I don't make any mistakes.

First, I know that to find f(-2), f(-1), and f(0), I need to substitute each of these x-values into the function f(x). So let me write down each one separately.

Starting with f(-2):

Let me compute the numerator and denominator separately to avoid mistakes. The numerator is 3*(-2) which is -6, then subtract 2, so -6 - 2 is -8. The denominator is (-2) - 2, which is -4. So putting it together, f(-2) is (-8)/(-4). Hmm, dividing two negatives gives a positive, so that's 2. Got that, f(-2) is 2.

Next, let me compute f(-1):

f(-1) = (3*(-1) - 2) / ((-1) - 2)

Now, f(0):f(0) = (3*0 - 2) / (0 - 2)

Compute numerator and denominator. Numerator: 3*0 is 0, minus 2 is -2. Denominator: 0 - 2 is -2. So f(0) is (-2)/(-2), which is 1. So f(0) is 1.Now, I need to add these three values together: f(-2) + f(-1) + f(0) which is 2 + (5/3) + 1. Hmm, let me convert all numbers to fractions to make it easier. 2 is equal to 6/3, 1 is equal to 3/3, and 5/3 is already a fraction. So adding them up:6/3 + 5/3 + 3/3.Adding the numerators: 6 + 5 + 3 is 14, so the total is 14/3.Wait, hold on, let me double-check my calculations because 2 + 5/3 + 1 is 3 + 5/3, which is 14/3. Yeah, that seems right.But let me make sure I didn't make any mistakes in computing each f(x). Maybe I should check each substitution again.Starting with f(-2):Numerator: 3*(-2) is indeed -6, minus 2 is -8. Denominator: (-2) - 2 is -4. So -8/-4 is 2. That seems correct.f(-1):Numerator: 3*(-1) is -3, minus 2 is -5. Denominator: (-1) - 2 is -3. So -5/-3 is 5/3. That's correct. f(0):Numerator: 3*0 is 0, minus 2 is -2. Denominator: 0 - 2 is -2. So -2/-2 is 1. Correct.Adding 2 + 5/3 + 1: 2 is 6/3, 5/3 is 5/3, and 1 is 3/3. So 6/3 + 5/3 + 3/3 is 14/3. That is 4 and 2/3. Hmm, I wonder if I can express that as a mixed number, but the question says to express as a common fraction, so 14/3 is the answer.Wait, but just to make sure I didn't make a mistake in the initial function. The function is f(x) = (3x - 2)/(x - 2). Yeah, that looks right. So for x = -2, x = -1, x = 0, I substituted correctly.Wait a second, but let me make sure about f(0). If I plug in x = 0, it's (0 - 2)/(0 - 2) which is (-2)/(-2) = 1. Yeah, that's correct.Wait, but just to make sure I didn't make any computation errors: 3*(-2) is -6, minus 2 is -8, divided by (-4) is 2. Correct.3*(-1) is -3, minus 2 is -5, divided by (-3) is 5/3. Correct.3*0 is 0, minus 2 is -2, divided by (-2) is 1. Correct.So adding them together: 2 is 6/3, 5/3 is 5/3, and 1 is 3/3. So 6/3 + 5/3 + 3/3 is 14/3. Yes, that seems right.Wait, but I just thought, is there any chance that x = 2 is a problem? Because the function is undefined at x = 2, but in this case, we're plugging in -2, -1, and 0, so x = 2 isn't an issue here. So all three f(x) are defined, and we can safely compute them. Therefore, the sum is 14/3.But just to be absolutely thorough, let me compute each term again:For f(-2): Numerator: 3*(-2) = -6, minus 2 is -8.Denominator: (-2) - 2 = -4.So -8 divided by -4 is 2. Correct.For f(-1): Numerator: 3*(-1) = -3, minus 2 is -5.Denominator: (-1) - 2 = -3.So -5 divided by -3 is 5/3. Correct.For f(0): Numerator: 3*0 is 0, minus 2 is -2.Denominator: 0 - 2 is -2.So -2 divided by -2 is 1. Correct.Adding them together: 2 + 5/3 + 1.2 + 5/3 + 1.2 is 6/3, 1 is 3/3, so 6/3 + 5/3 + 3/3 = (6 + 5 + 3)/3 = 14/3. So yes, 14/3 is the correct answer.Wait, but just to make sure, maybe I can represent 14/3 as a mixed number? 14 divided by 3 is 4 with a remainder of 2, so that's 4 and 2/3. But the question says to express as a common fraction, so 14/3 is fine.Alternatively, if I were to write it as a decimal, it's approximately 4.666..., but the question wants an exact fraction, so 14/3 is the way to go. I think that's thorough. I don't see any mistakes in my calculations, so I'm confident that 14/3 is the correct answer.**Final Answer**The value of $( f(-2) + f(-1) + f(0) )$ is $\boxed{\dfrac{14}{3}}$.

[2095 Tokens]

Okay, so I need to find f(-2) + f(-1) + f(0) where f(x) is given by (3x - 2)/(x - 2). Hmm, let's start by plugging in each value step by step.

First, f(-2):
f(-2) = (3*(-2) - 2)/(-2 - 2) = (-6 - 2)/(-4) = (-8)/(-4) = 2.
Next, f(-1):
f(-1) = (3*(-1) - 2)/(-1 - 2) = (-3 - 2)/(-3) = (-5)/(-3) = 5/3.
Then, f(0):
f(0) = (3*0 - 2)/(0 - 2) = (-2)/(-2) = 1.
Now, add them together: 2 + 5/3 + 1. That's 3 + 5/3.
Convert 3 to thirds: 9/3 + 5/3 = 14/3.

The value of f(-2) + f(-1) + f(0) is $\boxed{\dfrac{14}{3}}$.
To find $( f(-2) + f(-1) + f(0) )$ where $( f(x) = \frac{3x - 2}{x - 2} )$:
1. Calculate $( f(-2) )$:$[f(-2) = \frac{3(-2) - 2}{-2 - 2} = \frac{-6 - 2}{-4} = \frac{-8}{-4} = 2]$2. Calculate $( f(-1) )$:$[f(-1) = \frac{3(-1) - 2}{-1 - 2} = \frac{-3 - 2}{-3} = \frac{-5}{-3} = \frac{5}{3}]$3. Calculate $( f(0) )$:$[f(0) = \frac{3(0) - 2}{0 - 2} = \frac{-2}{-2} = 1]$Add the results together:$[2 + \frac{5}{3} + 1 = 3 + \frac{5}{3} = \frac{9}{3} + \frac{5}{3} = \frac{14}{3}]$The value is $(\boxed{\dfrac{14}{3}})$.

[567 Tokens]

Figure 17: Further example demonstrating improvements in reasoning approach

## O    LIMITATIONS

Despite our work's effective improvements in performance and efficiency, limitations remain. Our and most previous works focus primarily on the math as it provides an excellent verification environment and testbed for validating new methodologies. We believe further validation in code generation and agentic tasks would be valuable to determine if similar favorable trade-offs can be achieved in these contexts. Importantly, our methods were not specifically designed for mathematical tasks but were developed as domain-agnostic approaches that should naturally extend to other areas. In future work, we plan to explore more realistic scenario tasks, particularly those involving agentic reasoning, to further validate our approach and improve the efficacy-efficiency trade-off in more broader areas.

