# OpenReview forum: "Learn to Reason Efficiently with Adaptive Length-based Reward Shaping"
_ICLR.cc/2026/Conference — ICLR 2026 Poster_

### Official Review · Reviewer_2Nyo · 2025-10-30

**Soundness:** 3
**Presentation:** 4
**Contribution:** 3
**Rating:** 6
**Confidence:** 3

**Summary:**

This paper addresses the problem of inefficiency in Large Reasoning Models (LRMs) caused by over-thinking, where models generate unnecessarily long reasoning chains. The authors propose LASER and LASER-D, two adaptive length-based reward shaping methods that dynamically adjust reasoning length according to task difficulty. The approach provides a unified framework that formalizes existing CoT compression techniques and introduces a fully automated RL method that balances conciseness and reasoning depth. Experiments show significant improvements in both accuracy and token efficiency across multiple models and benchmarks.

**Strengths:**

1. The paper presents a unified conceptual framework that reformulates various length-control methods as instances of length-based reward shaping. This brings theoretical clarity to prior work and establishes a strong foundation for the proposed methods.

2. The design of LASER and LASER-D is elegant and well-motivated. The difficulty-aware mechanism that adjusts reasoning budget based on in-batch correctness is intuitive and largely automated, reducing the need for manual hyperparameter tuning.

3. Empirical results are very strong. LASER-D achieves large gains in both accuracy and efficiency (e.g., +5.3% accuracy with a 64% token reduction on AIME2024), demonstrating a clear improvement in the accuracy-efficiency trade-off.

4. The evaluation is thorough and convincing. The authors show consistent gains across model scales and domains, and the behavioral analysis offers meaningful insights into how the method reduces unproductive reasoning.

**Weaknesses:**

1. The difficulty estimation based on in-batch correctness is a noisy heuristic. It may misclassify easy problems as hard, leading to inefficient allocation of reasoning budget and potential instability during training.

2. The “automatic” mechanism introduces new hyperparameters (e.g., monitoring size, update frequency) whose tuning is not well discussed. This adds complexity and may limit practical applicability to new domains.

3. The analysis of reasoning behavior relies on keyword counting, which is a coarse proxy for reflective thinking. The model could learn to express similar unproductive reasoning with different phrasing, weakening the interpretability of this analysis.

4. The step-function reward creates a sharp incentive boundary at the target length. This may encourage minimal valid responses rather than richer reasoning, potentially sacrificing explanation quality in borderline cases.

**Questions:**

1. How stable is the in-batch difficulty estimation across training, and have you tried combining it with more static difficulty signals?

2. Did you observe cases where the model learned to bypass the reflection keyword filter using alternative phrasing?

---

> ### Author Response · Authors · 2025-11-25
> **Response to the Review (1/2)**
>
> Thanks for your insightful review and feedback.
>
> ### W1
> ```
> The difficulty estimation based on in-batch correctness is a noisy heuristic. It may misclassify easy problems as hard…
> ```
>
> Thanks for the question. Indeed, variance would exist in estimation methods. To validate the variance, we conducted an additional analysis: we randomly sampled 50 training questions and, for each question, generated 8 rollouts *eight independent times* (64 samples per question). We then measured the variance of the number of correct responses across these eight sampling groups. The results are shown below:
>
> | Model | Avg Variance |
> |-------|--------------|
> | DeepSeek-R1-Distill-Qwen-1.5B | 0.011831 |
> | + Laser-D Training | 0.012510 |
>
> The variance is very small, meaning the fluctuations in the number of correct rollouts are on average well below one response per question. This indicates that, in our experimental settings, the in-batch correctness signal is sufficiently stable for reliable difficulty estimation.
>
> ### W2
> ```
> The “automatic” mechanism introduces new hyperparameters (e.g., monitoring size, update frequency) whose tuning is not well discussed…
> ```
> Thanks for the suggestion. In our experiment, we did not tune monitoring size and update frequency and just set them as default values. To address the concerns, we further investigate the effect of the monitoring size and update frequency. Using Laser-D-L2048 on DeepSeek-R1-Distill-Qwen-1.5B (default: 500-sample monitoring set, update interval = 20), we conduct four additional runs by varying the update interval (10/30) and the monitoring set size (250/1000).
>
> As shown in the table below, all variants achieve far better effectiveness and efficiency than the original model. This demonstrates that our method is robust to these hyperparameter choices. More frequent updates or a larger monitoring set yield slight accuracy improvements. Overall, the default setting (interval = 20, monitoring size = 500) works well in practice and difference choices remain robust.
>
> |Method|MATH(Acc/Tokens)|AIME(Acc/Tokens)|AMC(Acc/Tokens)|OlympiadBench(Acc/Tokens)|Avg(Acc/Tokens)|
> |------|----------------|----------------|----------------|--------------------------|----------------|
> |Original|83.9 / 5042|28.9 / 15956|71.6 / 8202|43.3 / 11510|56.9 / 10177|
> |Laser-D-L2048|82.2 / 1623|31.0 / 5158|73.3 / 2572|46.2 / 2960|58.2 / 3059|
> |w Interval 10|83.1 / 1532|31.9 / 5325|73.8 / 2680|46.3 / 3015|58.8 / 3138|
> |w Interval 30|82.0 / 1850|30.8 / 5099|72.6 / 2465|45.7 / 2968|57.8 / 3095|
> |w 1000 Monitoring Set|83.1 / 1891|31.6 / 5204|73.3 / 2339|46.3 / 3033|58.6 / 3116|
> |w 250 Monitoring Set|82.8 / 1759|30.6 / 5473|72.5 / 2693|45.8 / 3051|57.9 / 3244|
>
>
> ### W3 & Q1
> ```
> The analysis of reasoning behavior relies on keyword counting, which is a coarse proxy for reflective thinking…
> ```
>
>
> Thanks for the comment. We agree that keyword counting alone can be a coarse proxy for reflective thinking. In fact, in our original submission, we would like to clarify that our analysis **does not rely solely on keyword statistics**. As explained in Line 469 - 476 in the original pdf, in addition to the keyword-based trends, we also perform a more fine-grained reasoning-behavior analysis using a proprietary LLM to identify behaviors such as backtracking, verification, enumeration, and subgoal setting, as well as detailed case studies. These analyses are presented in Section 6 and Appendix N of original/current PDF. We have now revised the PDF (changed highlighted with blue text) to make these components more prominent.
>
>
> ```
> Did you observe cases where the model learned to bypass the reflection keyword filter using alternative phrasing?
> ```
> Your concern is valid that keyword metrics may miss paraphrased forms of unproductive reasoning. However, as shown in Figure 4, even though *backtracking* (the keyword-tracked behavior) decreases substantially, other behaviors such as *verification*, *enumeration*, and *subgoal setting* remain stable or even increase. This indicates that the model is not merely rephrasing redundant reasoning, but is genuinely shifting toward more structured and efficient patterns of thought.
>
> Our qualitative case studies (Appendix N) further support this interpretation. After LASER-D training, the model tends to use cleaner, more formula-driven subgoal decompositions instead of verbose backtracking, suggesting a healthier and more principled reasoning style rather than superficial compression.

---

> ### Author Response · Authors · 2025-11-25
> **Response to the Review (2/2)**
>
> ### W4
> ```
> The step-function reward creates a sharp incentive boundary at the target length. This may encourage minimal valid responses…
> ```
> Thanks for the insightful question. We address the concern from four complementary perspectives.
>
> (1) The step-function reward only encourages reaching a target length, not producing the shortest possible chain. This keeps correctness and shaped reward aligned and avoids the degenerated “minimal response” behavior observed in some group-based methods. Evidence is given in Section 3.2, Figure 8(a)/(b), and Appendix E.
>
> (2) The difficulty-aware mechanism further prevents over-shortening on harder problems.
> LASER-D explicitly prioritizes correctness when a question is difficult. This design stabilizes training and avoids the “minimal valid response” failure mode.
>
> (3) Encouraging shorter *but equally correct* reasoning is the goal of efficient LRMs. Existing LRMs tend to produce highly redundant chains of thought; our method is designed to make reasoning more concise without sacrificing efficacy.
>
> (4) Empirically, explanation quality does not degrade; reasoning becomes healthier. As shown in Section 6 and the case studies in Appendix N, LASER-D does **not** reduce reasoning richness. Instead, the model tends to use clearer, richer, and healthier reasoning patterns.
>
>
> ### Q2
> ```
> Have you tried combining it with more static difficulty signals?
> ```
> Thanks for the question. Our method is motivated by the idea that difficulty should be modeled **dynamically**, because the difficulty a model experiences changes with its evolving reasoning capability during RL. Static difficulty annotations cannot capture this shifting landscape. The stability analysis in W1 shows that the dynamic signal is reliable in practice, making it well-suited for guiding adaptive target lengths.

---

### Official Review · Reviewer_MWVL · 2025-10-30

**Soundness:** 2
**Presentation:** 3
**Contribution:** 2
**Rating:** 4
**Confidence:** 4

**Summary:**

This paper proposes a simple framework, LASER, and an advanced version of i,t named LASER, which use reward shaping and dynamic reward shaping methods to do reinforcement learning, respectively, to improve the large reasoning models' efficiency and reasoning ability. The paper starts from a bunch of findings, which motivate their method contribution with difficulty dynamic reward shaping.

**Strengths:**

1. The paper conducts a lot of evaluation experiments, which point out the deficiency of the current reward shaping methods for efficient LRM training.
2. The design of the method, especially how to monitor the difficulty dymanics, and the logic of the paper are reasonable.
3. The experiment section includes a lot of baselines and different length settings, and different sizes of models.

**Weaknesses:**

1. While the method includes some novelty from the length penalty findings, it seems like an improved version of the previous method.
2. The task in the experiment is limited, which only contains mathematical reasoning.
3. The model family is limited, which only contains 2 versions of DeepSeek models.
4. While the tables in the experiment section show the effectiveness of this method, it is still a little bit hard to compare methods because sometimes other methods are better at token usage.

**Questions:**

1. Is it possible to include experiments with other model families, such as Qwen3 and LLama
2. Is it possible to include some experiments in other reasoning tasks, such as coding or commonsense reasoning?

---

> ### Author Response · Authors · 2025-11-25
> **Response to the Review (1/2)**
>
> Thanks for your insightful review and feedback.
>
> ### W1
> ```
> While the method includes some novelty from the length penalty findings, it seems like an improved version of the previous method.
> ```
>
> To clarify, our work goes beyond simply improving prior methods. We first revisit and unify previous approaches which systematically identify the core issues in RL for efficient reasoning, rather than providing an incremental refinement. Through this analysis, we uncover several previously overlooked limitations and introduce a new method that addresses them both conceptually and empirically.
>
> Our work contributes novel **dynamic** and **difficulty-aware** length adaptation designs, two aspects that have not been explored in prior efficient-reasoning methods.The resulting approach achieves significant empirical results, where we are able to reduce 64% token usage on hard questions like AIME while still achieving significant accuracy improvement (+5.3).
>
> ### W2 & Q1
> ```
> The task in the experiment is limited, which only contains mathematical reasoning
> ```
> We would like to remind the reviewer that  we have already included results on out-of-domain general-reasoning tasks (GPQA, LSAT, MMLU) and coding tasks (LiveCodeBench) in our original and current submission (section 5.4 and appendix C). Now we have updated the PDF to move the figures and tables to Section 5.4. For convenience, we report the numerical results below. We recommend checking the figures in the pdf, which provide a more intuitive comparison of the trade-off across methods.
>
> **General Reasoning**:
>
> | **Model / Dataset** | **GPQA** | **LSAT** | **MMLU** | **Across‑All (Avg.)** |
> |------------------|:-------------------:|:------------------:|:-----------------:|:------------------------:|
> | **Original** | 33.62 / 11200 | 24.62 / 15421 | 45.52 / 2691 | 34.59 / 9771 |
> | **Truncation‑4096** | 33.1 / 3182 | 24.9 / 3783 | 46.0 / 786 | 34.7 / 2580 |
> | **Truncation‑8192** | 33.4 / 3843 | 25.1 / 5605 | 46.3 / 930 | 34.9 / 3457 |
> | **Truncation‑10240** | 34.2 / 5480 | 25.0 / 7420 | 46.6 / 840 | 35.3 / 4585 |
> | **Group‑based-0.05** | 34.0 / 3426 | 23.6 / 3284 | 45.0 / 485 | 34.2 / 2393 |
> | **Group‑based‑0.1** | 34.2 / 3727 | 22.5 / 3810 | 44.5 / 668 | 33.7 / 2727 |
> | **Group‑based‑0.2** | 34.1 / 4229 | 22.2 / 5220 | 44.0 / 819 | 33.4 / 3407 |
> | **Laser‑L2048** | 35.1 / 4605 | 25.6 / 5912 | 46.6 / 650 | 35.8 / 3717 |
> | **Laser‑L4096** | 35.0 / 5202 | 25.2 / 6554 | 46.3 / 850 | 35.6 / 3556 |
> | **Laser‑L8192** | 35.0 / 6123 | 25.5 / 7886 | 45.9 / 1080 | 35.6 / 3683 |
> | **Laser‑D‑L1024** | 35.1 / 3956 | 25.2 / 6034 | 46.6 / 607 | 35.6 / 3576 |
> | **Laser‑D‑L2048** | 35.8 / 4577 | 25.3 / 6167 | 46.8 / 817 | 36.0 / 3845 |
> | **Laser‑D‑L4096** | 35.3 / 4856 | 25.0 / 7200 | 46.5 / 1000 | 35.6 / 3933 |
>
> **Coding Capbilities**:
>
> LiveCodeBench (Code Generation):
> | Model                     | Acc    | Tokens |
> |---------------------------|--------|--------|
> | Original                  | 24.091 | 13120  |
> | Laser-D   |
> | Laser-D-l2048-1.5b        | 28.409 | 6420   |
> | Laser-D-l4096-1.5b        | 28.409 | 6600   |
>
> As shown in the tables, our methods still achieve the best trade-off on the three out-of-domain general reasoning benchmarks, while the accuracy and efficiency are both improved significantly. For coding ability, our methods also achieve significant improvement while reducing the token usage by about 50%.
>
> ### W3 & Q2
> ```
> The model family is limited, which only contains 2 versions of DeepSeek models.
> ```
>  We clarify that our experiments contain more than 2 versions of DeepSeek models, and we
> respectfully disagree that our model family is limited. Concretely, our experiments are conducted across **five** different models, DeepSeek-R1-Distill-Qwen-1.5B/7B/32B, DeepSeek-R1-Distill-Llama-3.1-7B, and OpenReasoning-Nemotron-1.5B, which span multiple sizes and model families (see Tables 4 & 5, Section 5.3 in our original/current PDF). In the rebuttal phase, we also added more results on DeepScaleR-1.5B in Table 5 (current PDF). The results demonstrate our methods are still effective. Although DeepScaleR-1.5B is already trained with 3-stage RL, the model further trained by our approach demonstrates significantly better efficacy and efficiency.

---

> ### Author Response · Authors · 2025-11-25
> **Response to the Review (2/2)**
>
> ### W4
> ```
> While the tables in the experiment section show the effectiveness of this method, it is still a little bit hard to compare methods because sometimes other methods are better at token usage.
> ```
> Thanks for the question. As we note in the paper, efficient-reasoning methods involve an inherent **accuracy–efficiency trade-off**, so comparing token usage or accuracy in isolation can be misleading. Some baselines use fewer tokens on certain datasets, but often at an unacceptable accuracy cost—for example, L1-Max and Truncation reduce tokens on AIME but drop accuracy to ~20%. This is why we report trade-off plots and averaged metrics in the main paper.
> LASER-D offers the strongest overall balance, improving AIME accuracy by +5.3 while cutting token usage by roughly ~60%. We therefore encourage evaluating methods from the trade-off perspective rather than focusing solely on token counts.

---

### Official Review · Reviewer_k3oM · 2025-10-31

**Soundness:** 2
**Presentation:** 3
**Contribution:** 3
**Rating:** 6
**Confidence:** 4

**Summary:**

This paper addresses the "over-thinking" issue in Large Reasoning Models (LRMs), where models generate excessively long reasoning traces. The authors propose a unified framework for viewing various RL based chain-of-thought compression methods through the lens of length-based reward shaping. Building on this, they introduce two novel methods: LASER, which uses a step function to reward shorter correct responses, and LASER-D, which dynamically adapts the target length based on question difficulty and the model's evolving state during training. Experiments on several math reasoning benchmarks show that the proposed methods achieve significant improvements in token efficiency while also showing some improvements in reasoning accuracy compared to baselines.

**Strengths:**

1. The paper presents empirical results, demonstrating a strong trade-off between efficiency and performance. The proposed methods, consistently reduce token usage by a good margin across multiple models and benchmarks while often improving accuracy.
2. The introduction of a unified framework for RL-based CoT compression is a good conceptual contribution. It provides a clear and structured approach to understanding and comparing different reward shaping strategies.

**Weaknesses:**

1. Framing and Claims: The framing of performance improvements and baseline comparisons could be strengthened. The paper claims adaptive methods like AutoThink and Thinkless remain "verbose on hard ones (AIME)." However, the data in Table 4 shows these methods do reduce token usage on AIME compared to the original model, albeit less than LASER-D. Furthermore, AutoThink achieves higher accuracy on AIME than several LASER and LASER-D variants, which complicates the claim.
2. Deepseek-r1 distilled models are typically known to be very verbose shown in several recent works. Results on Qwen3 or even DeepScaleR would make the results more impactful.
3. For fair comparison, results should also be compared against officially released, downloadable models from prior work when available (e.g., L1-Max) that have the same starting model, since there may be small changes in training setup.
4. The unification framework is a strong point, but its coverage doesn't seem to be exhaustive. For instance, It is unclear how it would accommodate methods where a budget is enforced structurally, for example, by inserting an "end think" token after a certain number of steps (eg, in case of Qwen3).

**Questions:**

1. Could you clarify the choice of alpha=0.01 for the L1-Max baseline, as the original paper reports alpha=0.002? Was this value tuned, and if so, how?
2. Can you confirm that all reproduced baselines used the identical GRPO implementation and hyperparameters (including the "clip-higher" strategy), with the reward shaping term being the only difference?
3. How does the difficulty-aware mechanism handle the "cold-start" problem at the beginning of training, where many prompts may have zero correct rollouts, potentially making the difficulty estimation noisy? Maybe this is not an issue for the training dataset used, but could this be a limitation for training on harder datasets?

---

> ### Author Response · Authors · 2025-11-25
> **Response to the Review (1/2)**
>
> Thanks for your insightful review and feedback.
>
> ### W1
> ```
> Framing and Claims: …However, the data in Table 4 shows these methods do reduce token usage on AIME compared to the original model, … AutoThink achieves higher accuracy on AIME than several LASER and LASER-D variants, which complicates the claim.
> ```
>
> Thanks for the question. Our statement follows the fact shown in Table 4, these methods indeed reduce tokens relative to their original models but still consume around ~10k tokens on AIME (e.g., Thinkless reduces only ~33%). Under our definition, this length level of responses is still considered verbose. In contrast, LASER-D uses only ~5k–5.5k tokens (a 65–70% reduction).
> Regarding accuracy, we emphasize that efficacy and efficiency form an inherent trade-off. Thus, comparing accuracy alone or token usage alone may be misleading, which is also the reason we plot the figures (1/2/5/6) to compare the trade-off among different methods. While AutoThink achieves higher accuracy on AIME than some LASER/LASER-D variants, it does so by using substantially more tokens (9,514 vs. our 5,750). Moreover, its *overall* trade-off is less favorable: AutoThink attains a lower average score (57.6 vs. 60.3) while consuming far more tokens (5,581 vs. 3,520). This aligns with our broader conclusion that LASER-D provides a more favorable efficacy–efficiency balance across benchmarks.
>
> ### W2
>
> ```
> Deepseek-r1 distilled models are typically known to be very verbose shown in several recent works. Results on Qwen3 or even DeepScaleR would make the results more impactful.
> ```
>
> We appreciate your insightful suggestion. We further conduct experiments on DeepScaleR model with $L_T=4096$. And the results demonstrate our methods are still effective. Compared to the baseline model, which is already trained with 3-stage RL, the model further trained by our approach demonstrates significantly better efficacy and efficiency. We will add these additional results to the next revision of the paper.
>
> | Method     | AIME | Tok  | AMC  | Tok  | MATH500 | Tok  | Olympiad | Tok  | Avg   | Tok  |
> |------------|------|------|------|------|----------|------|----------|------|-------|------|
> | DeepScaleR-1.5B   | 40.3 | 9550 | 82.3 | 5182 | 87.3     | 3695 | 50.8     | 6628 | 65.18 | 6263 |
> | DeepScaleR-Laser-D-L4096 | 42.5 | 5802 | 83.3 | 3978 | 88.5     | 2100 | 52.8     | 3349 | 66.78 | 3807 |
>
> ### W3
> ```
> For fair comparison, results should also be compared against officially released, downloadable models from prior work when available (e.g., L1-Max) that have the same starting model, since there may be small changes in training setup.
> ```
> Thanks for the suggestion. We agree that comparing against officially released checkpoints is valuable when the starting models are aligned. In this case, however, the official L1-Max model is **not directly comparable** to our LASER/LASER-D models because it is trained on **DeepScaleR-1.5B**, which has undergone *three additional RL stages* beyond our base model DeepSeek-R1-Distill-Qwen-1.5B.
>
> Nevertheless, for completeness, we downloaded the official L1-Max checkpoint and evaluated it under our standardized evaluation setup:
>
> | model | MATH_ACC | MATH_TOK| AIME_ACC | AIME_TOK | AMC_ACC | AMC_TOK | OlympiadBench_ACC | OlympiadBench_TOK | AVG_ACC | AVG_TOK |
> |-------|-------|-------|--------|--------|--------|--------|--------|--------|---------|---------|
> | Laser-D-L2048     | 82.2 |1623 | 31.0 |5158 |73.3 |2572 | 46.2 |2960 |58.2 |3059 |
> | L1-Max-Official | 84.4 | 1332 | 26.7 | 1968 | 73.9 | 1587 | 45.0 | 1534 | 57.6 | 1605 |
>
> We observe two trends.
> 1. The official L1-Max model shows a similar phenomenon as our L1 variants: its accuracy noticeably decreases compared to its own base model (e.g., 65.18 → 57.68 for theirs; 56.9 → 51.4 in our paper), consistent with our results.
> 2. Even though L1-Max starts from a much stronger checkpoint (DeepScaleR-1.5B), our **LASER-D still achieves higher average accuracy** (60.3 vs. 57.6) while preserving strong efficiency.
>
> We believe these aligned trends further strengthen both the credibility of our implementation and the effectiveness of our proposed methods.
>
> ### W4
> ```
> It is unclear how it would accommodate methods where a budget is enforced structurally, for example, by inserting an "end think" …(e.g. Qwen3)
> ```
> Thanks for the question. As explained in our original submission (Line 83-96 & Line 108-125 Table 1 in the current version), there are 3 categories of efficient CoT. And end-think belongs to budget-based method, but we focus on single-stage, full compression as clarified in the submission. Such budget mechanisms operate differently and are largely **orthogonal** to our approaches, though they could potentially be combined in future work.

---

> ### Author Response · Authors · 2025-11-25
> **Response to the Review (2/2)**
>
> ### Q1
> ```
> Could you clarify the choice of alpha=0.01 for the L1-Max baseline, as the original paper reports alpha=0.002? Was this value tuned, and if so, how?
> ```
> Thanks for the question! We double-checked our implementation and confirmed that the actual training used $\alpha = 0.002$, fully consistent with the official L1-Max code. The $\alpha = 0.01$ shown in the submission was a typo inherited from the unused config file in the official repository. In the official implementation, this config value is never passed into L1-Max training, and our re-implementation preserved that structure. We apologize for the typo and have corrected it.
>
> ### Q2
> ```
> Can you confirm that all reproduced baselines used the identical GRPO implementation and hyperparameters…
> ```
>
> Yes, we compare all methods fairly using the identical GRPO implementation as the reward shaping term being the only difference.
>
> ### Q3
> ```
> How does the difficulty-aware mechanism handle the "cold-start" problem at the beginning of training, where many prompts may have zero correct rollouts...could this be a limitation for training on harder datasets?
> ```
> This is a good question. In our case, the base model (DeepSeek-R1-Distill-Qwen-1.5B) already produces at least one correct rollout for about 75% of the training questions, so the difficulty-aware mechanism receives informative signals from the beginning.
>
> For extremely hard datasets where most prompts initially yield zero correct rollouts, indeed the mentioned problem will arise, but it is a general challenge for RL, not specific to our method. Common remedies in the community, such as oversampling strategies [1] or applying SFT/middle-training before RL [2,3], can be directly combined with our approach.
>
> [1] Yu et al., DAPO: An Open-Source LLM Reinforcement Learning System at Scale, CoRR 2503.14476
>
> [2] Wang et al., OctoThinker: Mid-training Incentivizes Reinforcement Learning Scaling, CoRR 2506.20512
>
> [3] FAIR CodeGen Team, CWM: An Open-Weights LLM for Research on Code Generation with World Models, CoRR 2510.02387

---

### Official Review · Reviewer_7JFv · 2025-11-01

**Soundness:** 3
**Presentation:** 2
**Contribution:** 2
**Rating:** 6
**Confidence:** 3

**Summary:**

This paper **(LASER)** aims at addressing the _overthinking_ phenomenon in Large Reasoning Models (LRM) through a novel RL reward shaping method. Specifically, it first draws inspiration from truncation-based baseline methods and existing length rewards in efficient reasoning by proposing a step function reward. It then takes problem difficulty and the RL training dynamics into account by further introducing an adaptive module **(LASER-D)** that controls the reward shaping hyper-parameters on-the-fly. Experiment results on various in-domain and out-of-domain benchmarks demonstrate that LASER effectively reduces reasoning length with minimal influence on the accuracy.

**Strengths:**

- The paper is well motivated and shows a good understanding of existing methods in LLM efficient reasoning.
- The experimental study seems comprehensive (applying the method to 5 LLMs and evaluating on both in-domain and out-of-domain tasks).
- The paper is rigorous in terms of including important details like hyper-parameter choices and abundant ablation studies in the main text & appendix.

**Weaknesses:**

1. My main concern lies in the numerous *manual design choices* (often appearing as hyperparameters) despite the paper’s claim that most modules are *automatically adaptive*. For instance, the criterion for defining $L_A$ is set as the smallest value ensuring $ECR_d \geq 1$, where “1” represents the threshold for “at least one complete and correct response.” Why fix this threshold as a constant rather than make it depend on the rollout size $K$? Similarly, the lower bound length $L_T$ appears to be manually chosen, and the experiments suggest that its value is adjusted for different model (e.g., 4096 for the 8B model and 8192 for the 1.5B model). Other examples of such manual tuning include the choice of **three** difficulty levels and the update interval of $N$ steps. For practitioners, what concrete guidance is provided for selecting these parameters in new settings?


2. While the overall presentation is clear, I find the explanation in Section 4.2 (*Automatic Adapting Mechanism*) particularly difficult to follow. In particular, the term “group” is confusing, as it can refer to both the “rollout group” (i.e., the $K$ rollouts for a given prompt) and the “difficulty group.” As a result, any statement that simply mentions “group” without specification becomes ambiguous. Additionally, the concept of $ECR_d$ seems to represent a per-difficulty quantity, yet its introduction follows immediately after the sentence “we sample $K$ responses for each query in the monitoring set, and $ECR$ is computed as…,” which makes it sound like a per-query statistic instead. I recommend a thorough revision of this subsection to improve clarity and precision.

3. [Minor] Based on Table 5, I notice that while LASER does work on non-Qwen models as well, the magnitude of length reduction is perceivably smaller than the two Qwen models highlighted throughout main text (on most benchmarks the reductions are $> 50$% on Qwen models, but only around $40$% for other models). Is there any intuitive reason that can explain such discrepancy?

**Questions:**

Most questions are raised in the weaknesses. Other than above:

1. [Question] During inference time, does LASER further uses the difficulty assessment module to evaluate the difficulty of a new query, or simply let the RL-finetuned model to generate the reasoning without any pre/post-processing?

2. [Typo] **L144:** lack of a verb after "In this section, we []".

---

> ### Author Response · Authors · 2025-11-25
> **Response to the Review (1/2)**
>
> Thanks for your insightful review and feedback.
>
> ### W1
>  ```
> My main concern lies in the numerous manual design…For instance, the criterion for defining $L_A$ is set as the smallest value ensuring $ECR_d \geq 1$...Why fix this threshold…the lower bound length  appears to be manually chosen, and the experiments suggest that its value is adjusted for different model…three difficulty levels and the update interval of  steps…
> ```
> Thanks for your question. **For all the mentioned hyperparameters, we emphasize that we never tune them in our experiments and instead keep them as reasonable values and fix them for simplicity**. Below, we explain the choice of these configurations one by one.
>
> **ECR Threshold**
>
> Overall, our ECR criterion is designed as a dynamic pass@k-style metric under a length constraint. As a straightforward choice and minimal requirement for RL, we choose “1” as the parameter. We set it as a constant for simplicity. Empirically, in our ablations (Table 10), varying the ECR threshold also produces minimal changes in accuracy and length. Thus, fixing the threshold to 1 offers a simple and stable mechanism.
>
> **Lower Bound for Target Length**
>
> First, we clarify our settings:
>
> - For **DeepSeek-R1-Distill-Qwen-1.5B**, we evaluate \(L_T = 1024/2048/4096\).
> - For **OpenReasoning-Nemotron-1.5B**, we use \(L_T = 8192\) because this model’s *original* output length (>20k tokens) is significantly larger than other 1.5B/7B models.
>
> Since $L_T$ serves as the minimal candidate target length for $L_A$. It influences the model’s position on the efficacy–efficiency trade-off observed in Figure 1 and Table 4: smaller values promote stronger compression, whereas larger values lead to more conservative length reduction. As a result, the choice of \(L_T\) is fundamentally about how users want to navigate the trade-off.
>
> - If one cares more about accuracy, a larger $L_T$ (e.g., ~$\frac{1}{3}$ of the context window) is appropriate.
> - If one prioritizes efficiency, a smaller $L_T$ encourages more aggressive compression.
>
> **Difficulty levels**
> Setting 3 difficulty levels is a common choice in prior works, such as [1-3]. And we follow this setting for simplicity.
>
> **Update interval**
> We didn’t really tune the update interval values, but to address the concern about effect of update intervals, we experiment with update intervals of **10, 20 (default), and 30**. The results show that our approach is robust. While more frequent update shows slightly better accuracy, all results are significantly better than baselines in both efficiency and efficacy.
>
> | Method            | MATH | Tok | AIME | Tok | AMC | Tok | Olympiad | Tok | Avg | Tok |
> |-------------------|------|-----|------|-----|-----|-----|----------|-----|-----|-----|
> | Laser-D-L2048     | 82.2 |1623 | 31.0 |5158 |73.3 |2572 | 46.2     |2960 |58.2 |3059 |
> | Interval 10       | 83.1 |1532 | 31.9 |5325 |73.8 |2680 | 46.3     |3015 |58.8 |3138 |
> | Interval 30       | 82.0 |1850 | 30.8 |5099 |72.6 |2465 | 45.7     |2968 |57.8 |3095 |
>
> [1] Zeng et al., SimpleRL-Zoo: Investigating and Taming Zero Reinforcement Learning for Open Base Models in the Wild COLM2025
> [2] Pikus et al., Hard Examples Are All You Need: Maximizing GRPO Post-Training Under Annotation Budgets arxiv. 2508.14094
> [3] Setlur et al., e3: Learning to Explore Enables Extrapolation of Test-Time Compute for LLMs arxiv. 2506.09026
>
>
> ### W2
>
> ```
> While the overall presentation is clear, I find the explanation in Section 4.2 (Automatic Adapting Mechanism)...I recommend a thorough revision of this subsection…
> ```
>
> We appreciate your helpful comments. We have updated the PDF to revise  this subsection to remove the ambiguities you mentioned.
>
> In the updated version, we (1) avoid the ambiguous use of the term “group,” (2) clearly state that $K$ rollouts are sampled *per query*, and (3) explicitly define $ECR_d$ as an aggregated statistic computed *over all monitoring queries with difficulty label $d$*. We have also reorganized the subsection so that difficulty assignment, coverage computation, and the selection of the adaptive target length are presented in a clear, sequential manner.
>
> We kindly invite you to check the revised PDF, where the updated formulation and notation are highlighted.

---

> ### Author Response · Authors · 2025-11-25
> **Response to the Review (2/2)**
>
> ### W3
> ```
> [Minor] Based on Table 5, I notice that while LASER does work on non-Qwen models as well, the magnitude of length reduction is perceivably smaller than the two Qwen models highlighted throughout main text…any intuitive reason that can explain such discrepancy?
> ```
>
> The discrepancy is an interesting observation. We believe there are some intuitive factors.
>
> 1. Different base models exhibit different reasoning-length behaviors before training. As shown in Table 5, Qwen-based models generally produce longer chains of thought by default, leaving more “headroom” for length reduction.
>
> 2. The hyperparameters and training recipes commonly used in the community, including those adopted in our work, have been tuned more extensively on Qwen models. As a result, Qwen models may benefit more directly from the current default settings.
>
> ### Q1
> ```
> [Question] During inference time, does LASER further uses the difficulty assessment module to evaluate…generate the reasoning without any pre/post-processing?
> ```
> Thanks for your question. Our methods are only used in the training phase. After training, models just generate texts without any pre-/post-processing.

---

> > ### Comment · Reviewer_7JFv · 2025-11-28
> > **Thanks for the clarification**
> >
> > Dear authors,
> >
> > Thanks for giving clarifications/answers to the questions I've raised. I also appreciate the polish to **Sec 4.2** following my comment in **W2** -- the current form looks much cleaner and easy to understand (despite one new question I'm having, see below).
> >
> > My questions are mostly addressed, so I'm glad to _raise my score_ in recognition of the work and the followup rebuttals. However, it seems that the recent "accident" makes ICLR disable the "edit" functionality. I will do this later when possible.
> >
> > _Remaining question:_
> >
> > Again for Sec 4.2, now I understand better that $ECR_d(l)$ tries to approximate an "expected number complete, correct responses at difficulty $d$ with length limit $l$", and $ECR_d(L_A) \approx 1$ _by construction_.
> >
> > However, I'm a bit confused about the definition of $P_{l, d}$: conditioning on $d$ we have that an expected $C_d$ samples (out of $K = 8$) will be _correct_; then for the _complete_ (under constraint $l$) part, shouldn't $P_{l, d}$ be defined as  fraction of all **correct** rollouts that are under length $l$? I believe such conditioning on correctness might be necessary for the "expectation" to work -- otherwise we are assuming that the probability of a rollout being correct is independent of it being complete (i.e. having length $\leq l$).
> >
> > I see that empirically the current choice of $P_{l, d}$ (a fraction out of all rollouts) can work just well, it's just there seems to be a gap between the "expectation analogy" and what is actually being implemented.
> >
> > Best,
> > Reviewer 7JFv

---

> > > ### Author Response · Authors · 2025-11-30
> > > **Thanks for the recognition**
> > >
> > > We appreciate the reviewer's recognition and this insightful question. Conceptually, we agree that if one wants $ECR_d(l)$ to *exactly* match the expected number of complete and correct rollouts, then $P_{l,d}$ could be defined over **correct** rollouts only. We see this conditional version as less biased in a probabilistic sense, while the unconditional version trades a bit of bias for lower variance. In practice, we considered both choices and their trade-offs. For easy questions, where most rollouts are already correct, the two versions are almost identical, so the impact is negligible. For hard questions, however, there are often only 1–2 correct rollouts per query, making the conditional ratio high-variance and potentially leading to unstable adaptation of $L_A$.
> > >
> > > From this perspective, we adopt the unconditional version as a simple, low-variance surrogate that only needs to be monotone in the length limit. Empirically, this choice leads to stable adaptation and strong performance. We will clarify this simplicity-variance trade-off in the revised version.

---

### Meta-Review · Area_Chair_CXLh · 2026-01-02

**Summary:**

This paper proposes LASER and LASER-D, two reinforcement learning–based methods for improving reasoning efficiency in large reasoning models via adaptive length-based reward shaping. The paper presents a unified perspective that formulates existing efficient-reasoning approaches under a length-based reward framework, and introduces a step-function reward that is further extended to be dynamic and difficulty-aware.

The paper received three borderline accepts and one borderline reject. While some reviewers raised concerns regarding the incremental nature of the algorithmic contribution and certain design choices, the rebuttal addressed most technical questions and strengthened the empirical evidence. Overall, the work presents solid empirical results on an important problem and merits acceptance as a poster.

**Reviewer Concerns:**

Reviewer 7JFv mainly raised concerns about the clarity of the automatic adapting mechanism and the number of manual design choices. The rebuttal substantially improved the presentation of Section 4.2 and provided robustness analyses for key hyperparameters. These concerns were largely addressed, and the reviewer increased their score.

Reviewer k3oM questioned the claims, baseline fairness, and whether the unified framework fully captures existing approaches. The rebuttal clarified comparisons, added evaluations on additional models, and explained design choices. These concerns were largely addressed.

Reviewer MWVL raised concerns about limited novelty, task coverage, and model diversity, viewing the method as an improved variant of prior length-penalty approaches. The rebuttal clarified that additional models and out-of-domain tasks were already included and highlighted the contributions of dynamic and difficulty-aware length adaptation designs. These concerns were largely addressed.

Reviewer 2Nyo expressed concerns about the noisiness of difficulty estimation, additional hyperparameters introduced by the “automatic” mechanism, and the sharpness of the step-function reward. The rebuttal provided variance analysis, robustness experiments, and further qualitative justification, addressing most of these concerns.

**Reviewer Scores:**

Reviewer 7JFv increased their score after the rebuttal and would likely maintain an accept, as their concerns were largely resolved.

Reviewer k3oM would likely maintain a borderline accept score, as their concerns were largely resolved.

Reviewer MWVL would likely increase the score to borderline accept, as concerns about incremental novelty, task coverage, and model diversity are resolved.

Reviewer 2Nyo would likely maintain a borderline accept, given that most technical concerns were addressed in the rebuttal.

---

### Decision · Program_Chairs · 2026-01-26

Accept (Poster)